# NORMALIZED ATTENTION WITHOUT PROBABILITY CAGE

## ABSTRACT

Despite the popularity of attention based architectures like Transformers, the geometrical implications of $\mathrm{softmax}$-attention remain largely unexplored. In this work we highlight the limitations of constraining attention weights to the probability simplex and the resulting convex hull of value vectors. We show that Transformers are biased towards local information at initialization and sensitive to hyperparameters, contrast attention to max- and sum-pooling and show the performance implications of different architectures with respect to biases in the data. Finally, we propose to replace the $\mathrm{softmax}$ in self-attention with normalization, resulting in a generally applicable architecture that is robust to hyperparameters and biases in the data. We support our insights with empirical results from more than 30,000 trained models. Implementations are in the supplementary material.

## 1 INTRODUCTION

The concept of neural attention (Graves, 2013; Bahdanau et al., 2015) has sparked a number of architectural breakthroughs. The Transformer architecture (Vaswani et al., 2017) successfully deploys multi-headed self-attention in several consecutive layers for natural language processing (NLP) – a solution that has become increasingly popular (Vaswani et al., 2017; Radford et al., 2018; 2019; Brown et al., 2020; Devlin et al., 2019; Yang et al., 2019; Raffel et al., 2019; Liu et al., 2019). Apart from NLP, self-attention has shown success in applications ranging from image classification (Parmar et al., 2019; Dosovitskiy et al., 2021) to generative adversarial networks (Zhang et al., 2019) to reinforcement learning (Bram et al., 2019; Parisotto et al., 2019; Loynd et al., 2020). The attention architecture choice is thereby often based on one, if

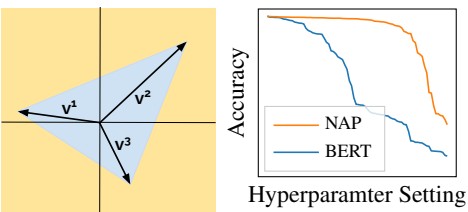

Figure 1: Softmax attention outputs can only lie within the convex hull spanned by the value vectors $\boldsymbol{v}^i$ (blue region). Removing this constraint with our normalized attention pooling (NAP) leads to an increased robustness to hyperparameters (See Section 5.1.1.)

not both, of the following arguments: (1) Attention helps with the information flow by providing more direct, dynamic links between inputs and outputs. (2) Attention is directly interpretable as one can investigate the percentages to which different inputs are "attended" to. However, this second argument has been challenged recently, as several works show that attention weights do not directly correlate with predictions (Jain & Wallace, 2019; Wiegreffe & Pinter, 2019; Pruthi et al., 2019; Brunner et al., 2020; Pascual et al., 2020). With interpretability already in dispute, we focus on the other argument. Specifically, we question whether attention is the best way to route information and argue that apart from the proven success, $\mathrm{softmax}$ attention might not always be the best option.

In this work, we investigate the theoretical implications of constraining the attention weights to the probability simplex, and propose an unconstrained alternative based on normalization. We show that the popular Transformer architecture has an innate bias towards local information at initialization and showcase implications of this architectural bias on biases in the data. We investigate different architectures on an abstract level in numerous experiments and demonstrate the advantage of unconstrained attention. In particular, we improve robustness to hyperparameters and show the general applicability of attention based architectures as compared to sum and max pooling.

## 2    BACKGROUND AND RELATED WORK

Many data processing tasks can be addressed by representing the input as a set or sequence of discrete tokens, e.g., words in a sentence, frames in a video or points in a point cloud. As a general formulation, we represent each input token by a vector $\boldsymbol{x}^i \in \mathbb{R}^d$ for $i \in \{1, \ldots, N\}$, where $N$ is the sequence length and $d$ is the dimensionality of each token. For ease of notation we use the word "sequence" throughout, but note that all architectures discussed are also applicable to unordered sequences, i.e., sets of tokens. Multi-headed dot-product self-attention is a fundamental building block of the Transformer architecture (Vaswani et al., 2017). It allows for information exchange between different tokens of the input sequence. More formally, for each attention head $m$, the input vectors $\boldsymbol{x}^i$ are projected by an affine transformation to a query $\boldsymbol{q}_m^i$, key $\boldsymbol{k}_m^i$ and value vector $\boldsymbol{v}_m^i$. The dimensionality of these vectors is chosen as $d_h = \frac{d}{M}$, where $M$ is the number of attention heads. The query and key vectors are used for a pairwise dot product, scaled by the square root of the head dimension $d_h$, to form the attention logits $l_m^{i,j} = \frac{<\boldsymbol{q}_m^i, \boldsymbol{k}_m^j>}{\sqrt{d_h}}$ and attention vectors $\boldsymbol{a}_m^i = \text{softmax}([l_m^{i,1}, \ldots, l_m^{i,N}])$, where $\text{softmax}$ refers to the normalized exponential function $\text{softmax}(\boldsymbol{x})^j = \frac{\exp(x^j)}{\sum_k \exp(x^k)}$ commonly used to project vectors to the probability simplex $\mathcal{S}_P = \{\boldsymbol{a}_m^i | a_m^{i,j} \geq 0 \; \forall j \text{ and } \sum_j a_m^{i,j} = 1\}$. The output $\boldsymbol{o}_m^i$ of each attention head $m$ is then given by a weighted sum of all value vectors $\boldsymbol{o}_m^i = \sum_j a_m^{i,j} \cdot \boldsymbol{v}_m^j$. These attention head outputs are concatenated and mixed trough an affine transformation to form the attention output in the Transformer architecture (Vaswani et al., 2017).

In this work, we investigate whether constraining the attention vectors $\boldsymbol{a}_m^i$ into the probability simplex through the $\text{softmax}$ function is sensible from a geometric perspective. We contrast the multi-head self-attention architecture to attention-inspired architectures without $\text{softmax}$ (discussed in Section 4) as well as simpler aggregation methods commonly used. Specifically, while Yun et al. (2020) show that Transformers are universal sequence-to-sequence function approximators, we question the practical necessity of an attention architecture, when sum pooling (Zaheer et al., 2017) already provides general function approximation capabilities (Zaheer et al., 2017; Xu et al., 2019; Segol & Lipman, 2020). Further, we compare to max pooling, a common aggregator choice that has shown good empirical success (Nagi et al., 2011; Zaheer et al., 2017; Velikovi et al., 2020). Several works have proposed architectural changes to the Transformer (Yun et al., 2020; Wu et al., 2019; Lan et al., 2020; Dehghani et al., 2019; Fan et al., 2020; Wu* et al., 2020; Press et al., 2019; Bachlechner et al., 2020), however, these do not alter the multi-head self-attention. Another direction of research has focused on improving the computational efficiency of Transformers (Wang et al., 2020; Lee et al., 2019; Katharopoulos et al., 2020; Zaheer et al., 2020; Choromanski et al., 2021). However, these either retain the softmax (Lee et al., 2019; Wang et al., 2020; Zaheer et al., 2020) or try to stay close to it (Katharopoulos et al., 2020; Choromanski et al., 2021; Lu et al., 2021). We refer to Tay et al. (2020) for a more extensive overview on works in this direction. Our focus is different: we ask whether the softmax is necessary at all. The Synthesizer (Tay et al., 2021) and MLP-Mixer (Tolstikhin et al., 2021) show that an aggregation with learned aggregation weights can already lead to good results. However, these aggregation schemes make the models dependent on the input sequence length – a trait we explicitly seek to avoid. Recently, a few works provide complementary results that motivate architectures without softmax. In particular, Tay et al. (2019) introduce an attention mechanism that allows for negative attention values. Schlag et al. (2021) link the head dimension $d_h$ to a corresponding limitation in the memorization capacity of linearized models. Shen et al. (2021) propose an architecture similar to our NON architecture (see below) for improved computational complexity. Cao (2021) also explore an architecture similar to our NON architecture and links it to a Petrov-Galerkin projection for operator approximation. They empirically find in an ablation that a layer normalization on the keys and queries separately improves convergence. Our work goes beyond this, showing the correlated effects between hyperparameters and architecture choices. In contrast to all mentioned works, we provide a viewpoint from the gradient dynamics perspective and investigate the effect of skewed data distributions.

## 3    IMPLICATIONS OF SOFTMAX ATTENTION

To start our discussion, we highlight an observation that follows directly from attention vectors $\boldsymbol{a}^i$ being constrained to the probability simplex $\mathcal{S}_P$:

**Outputs $o_m^i$ are convex combinations of the vectors $v_m^i$:** This in itself has interesting implications. First and foremost, we note that a convex combination of vectors $v_m^i$ cannot yield any vector outside the convex hull spanned by the value vectors $v_m^i$. An illustration of this output "cage" is given in Figure 1 (left). We conjecture that this constraint limits the hypothesis space of the optimization, making it more difficult for gradient descent to find a good solution. This conjecture is supported by our experimental results showing an increased robustness to hyperparameter choices when the constraint is removed. Further, we note the following from a theoretical perspective:

**No convex combination can in itself represent XOR:** A formal proof is given in Appendix A. Note that this implication highlights an inability to represent non-linearity. While the binary exclusive OR can be represented in architectures with multiple heads and layers, the insight further underlines our argument: An aggregation with weights constrained to the probability simplex is restrictive. The convex hull view point also leads to an additional insight:

**Transformers are not aggregation size independent:** Specifically, Transformers focus on local information at initialization; and this focus depends on the sequence length $N$. To see this, consider the embeddings $e^i$ after the first residual connection given by $e^i = x^i + W \left[ \sum_j a_1^{i,j} \cdot v_1^j \middle| \ldots \middle| \sum_j a_M^{i,j} \cdot v_M^j \right] + b$, where $[\cdot|\cdot]$ denotes concatenation and $W$ and $b$ represent the parameters of the affine transformation that mixes the $M$ attention head outputs to form the attention layer output.

Our aim is to show how much the embedding $e^i$ is influenced by the local information $x^i$ relative to the context information $\{x^j | j \neq i\}$. We first note that the contribution of context information depends on the initialization of $W$, where a typical initialization in language models yields a contractive projection.[1] This favors the residual connection, i.e., local information. However, even if we consider $W$ as scale preserving, we note that the magnitudes of the attention head outputs $o_m$ are upper bounded by the magnitudes of the value vectors $v_m$ as a result of the convex hull. Moreover, attention logits are normally close to 0 at initialization (to have the softmax in the unsaturated region). This yields attention to be close to mean aggregation as $o_m^i \approx \bar{v}_m = \frac{1}{N} \sum_j v_m^j$. Considering $v_m^j$ to be distributed

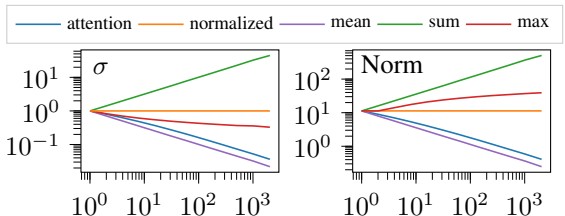

Figure 2: The standard deviation ($\sigma$) and norm of a pooling output depends on the sequence length $N$ (x-axis) if the output is not normalized. Softmax attention outputs (blue) scale similar to mean pooling at initialization, i.e., Transformers focus more on local information in longer sequences. Experiment details are given in Appendix C.

with zero-mean and variance $\sigma_v^2$ at initialization, we note that the expected magnitude and standard deviation of $\bar{v}_m$ scales proportional to $\frac{\sigma_v}{\sqrt{N}}$. This means that the fraction of context information in $e^i$ is dependent on the sequence length $N$ and, counterintuitively, is smaller for longer sequences. Specifically, with the constant contribution of the residual connection, Transformers focus more on local information; and more so in longer sequences than in shorter sequences. For reference, we visualize the dependence of $o_m$ on $N$ at initialization for different aggregators in Figure 2. We note that while an architectural bias towards local information might be beneficial in some applications, the implicit dependence on aggregation size $N$ is questionable. Finally, we note that:

**A softmax can easily saturate, which leads to vanishing gradients:** To show this, we look at the term $g = \frac{\delta a_m^{i,j}}{\delta l_m^{i,j}}$ which gives a gradient factor on the back-propagation path through the softmax. Specifically, we note that $g \in (0, 0.25]$ and that the range of $l_m^{i,j}$ for which $g > 0.01$ is less than 9.2. This means that any gradient update that changes $l_m^{i,j}$ by a constant of $c > 9.2$, e.g. by updating the bias terms, will most certainly lead to vanishing gradients henceforth. This is particularly worrisome in the Transformer architecture, where gradients correlate with the input due to the multiplicative interactions in the attention. This means that a single large input can result in a parameter update that yields vanishing gradients on all following data points. See Appendix B for proofs.

---

[1]As an example, BERT (Devlin et al., 2019) initializes $W$ with parameters drawn from a truncated normal distribution with standard deviation set to 0.02. This would only be roughly scale preserving for $d = 2500$.

## 4 NORMALIZED ATTENTION POOLING

Given the implications of $\mathrm{softmax}$ attention, one might seek alternatives. To investigate different architectural choices, we contrast the following architectures in an abstract empirical study:

**Transformer Encoder (BERT):** As a starting point, we replicate the encoder architecture presented by Vaswani et al. (2017) as described in the code release of Devlin et al. (2019).[2] This architecture has been used extensively (Radford et al., 2018; Yang et al., 2019; Raffel et al., 2019; Liu et al., 2019; Clark et al., 2020). Each Transformer-layer consists of two sub-modules: a multi-head self-attention "layer" and a feed forward network. Both modules are encompassed by residual connections. The multi-head self-attention "layer" consists of a

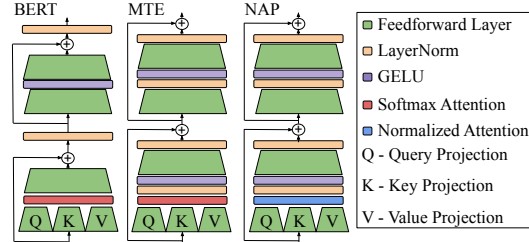

Figure 3: Difference in 1 Transformer layer.

projection to queries, keys and values, the attention mechanism as well as a mixing layer as described in Section 2. The feed forward network consists of two layers with a GELU (Hendrycks & Gimpel, 2016) non-linearity. Layer normalization (Ba et al., 2016) is applied *between* incoming and outgoing residual connections. Note that this gives a crucial distinction of this architecture: Embeddings are normalized *after* they are summed with the residual connection. This yields the implicit dependence on sequence length as discussed in Section 3. We train this architecture with learning rate warm-up and gradient norm clipping (cf. Devlin et al. (2019)).

**Modified Transformer Encoder (MTE):** To overcome the implicit dependence on sequence length, we modify the architecture by moving the layer normalizations and adding additional normalizations as shown in Figure 3. Note that this is different from the recently studied PreNorm (Parisotto et al., 2019; Nguyen & Salazar, 2019; Liu et al., 2020) that places the normalization before the attention mechanism. To further remove cofounding factors, we remove learning rate warm-up and gradient clipping, but keep a linearly decreasing learning rate schedule, taking Li et al. (2020) as reference. We provide an ablation of all modifications in Appendix D.1. All following architectures apply the same modifications. The resulting *MTE* architecture still projects attention weights to the probability simplex in the multi-head attention and is thereby limited to convex combinations of value vectors.

**Normalized Attention Pooling (NAP):** Given the success of online normalization during training - be it through batch- (Ioffe & Szegedy, 2015), layer- (Ba et al., 2016) , group- (Wu & He, 2020), instance- (Ulyanov et al., 2016) or weight-normalization (Salimans & Kingma, 2016) - our main proposal is to simply replace the softmax with a normalization:

$$\boldsymbol{a}_m^i = \mathrm{normalize}([l_m^{i,1}, \dots, l_m^{i,N}]) \qquad \text{with } \mathrm{normalize}(\boldsymbol{x})^j = g \cdot \frac{x^j - \mu_{\boldsymbol{x}}}{\sigma_{\boldsymbol{x}}} + b$$

where $\mu_{\boldsymbol{x}} = \frac{1}{N} \sum_j x^j$ and $\sigma_{\boldsymbol{x}} = \sqrt{\frac{1}{N} \sum_j (x^j - \mu_{\boldsymbol{x}})^2}$ are the mean and standard deviation of the corresponding input vector $\boldsymbol{x}$, in our case the logit vector calculated through key-query dot products. Similar to layer normalization (Ba et al., 2016), we introduce gain and bias parameters $g$ and $b$ initialized to 1 and 0, respectively. However, while Ba et al. (2016) introduce gain and bias vectors, we only introduce scalar parameters and broadcast these over the sequence, as we want the architecture to be independent of the sequence length $N$. Note that while no convex combination can represent the logical XOR, a normalized weighting can – we give a proof in Appendix A.

**No Online Logit Normalization (NON):** To investigate whether a dynamic normalization of the attention logits is necessary, we also train a model where we use the logits $l_m^{i,j}$ directly as attention weights, i.e., $\boldsymbol{o}_m^i = \mathrm{GELU}(\frac{1}{\sqrt{N}} \sum_j l_m^{i,j} \cdot \boldsymbol{v}_m^j)$. The factor $\frac{1}{\sqrt{N}}$ is introduced such that the model has in expectation a constant contribution of context at initialization.

**Simple Summation of Embeddings (sum):** From a theoretical perspective, summation is sufficient for general function approximation (Zaheer et al., 2017; Xu et al., 2019; Segol & Lipman, 2020). Therefore, we investigate to simply replace attention with a sum-reduce-broadcast operation.

---

[2]https://github.com/google-research/bert

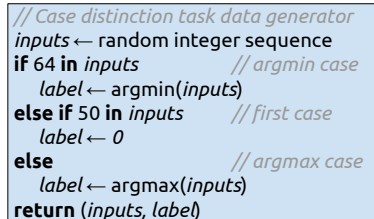
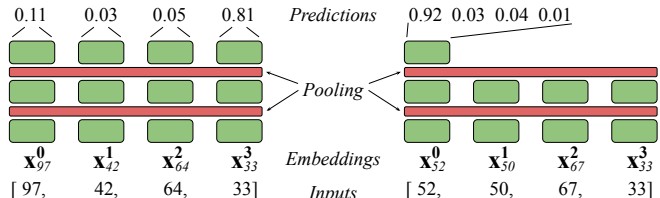

Figure 4: **Left:** Pseudo code of the data generation. The case distinction points 64 and 50 are chosen arbitrarily. **Right:** Task setup for outputs across all tokens (cf. Section 5.1) and outputs from the first token (cf. Section 5.2). Green boxes represent the trainable network layers (shared across tokens) while red boxes represent the pooling across tokens. The targets of the displayed examples would be [0, 0, 0, 1] and [1, 0, 0, 0], respectively.

**Max Pooling over Sequence Dimension (max):** Similar to sum pooling, we can replace the attention sub-module with a simple max-reduce-broadcast operation over the sequence dimension. Note that max pooling is a powerful operation that yields a direct link to up to $d$ different tokens.

We provide a schematic figure of all architectures in Appendix D. If not varied in a corresponding experiment, we default architecture hyperparameters to $L = 2$ Transformer-layers (consisting of an attention sub-module and feed forward sub-module each), $M = 4$ heads to calculate the logits (if applicable), $d = 128$ as model dimension and train on a total of 3200 batches of 32 example sequences each, using the Adam optimizer (Kingma & Ba, 2014). The hidden dimension of the feed forward sub-modules is $4 \cdot d$ for the models *BERT*, *MTE*, *NAP* and *NON*. For the models *sum* and *max* we increase the feed forward hidden dimension to approximately match the parameter count.

## 5 EXPERIMENTS AND RESULTS

Our goal with this work is to provide an insight into the variety of performance implications that the architecture choices entail. We aim to provide these insights independent of any particular application, as these architectures can be applied to a variety of tasks – from NLP (Vaswani et al., 2017; Devlin et al., 2019) to graph neural networks (Velickovic et al., 2018; Yun et al., 2019; Xu et al., 2019; Velikovi et al., 2020) to reinforcement learning agents (Parisotto et al., 2019; Fang et al., 2019; Loynd et al., 2020). We therefore focus on carefully crafted synthetic tasks that (1) are general enough in that we can expect the insights to generalize to a large set of downstream tasks and (2) let us modify key aspects that are hidden in real world data sets, such as a bias towards a certain sub-task. The focus on synthetic tasks also allows us to get a better grasp on the learning dynamics – the focus of this work – as we can train thousands of models in diverse hyperparameter combinations. To limit the influence of confounding hyperparameters, we generate new data points for every batch. This allows us to omit regularizations and their confounding factors. See Appendix D.2 for an in depth discussion of this setup as well as regularized ablations.

### 5.1 ARGMIN-FIRST-ARGMAX CASE DISTINCTION TASK

As a first task, we train the networks to pin-point a specific, input dependent token. Note that the ability to pin-point a specific token is an abstract task relevant to NLP (e.g., question answering or co-reference resolution), graph neural networks (e.g., finding the next hop in a shortest path) as well as reinforcement learning (e.g., action credit assignment). We consider an input pipeline where tokens from a fixed integer-vocabulary are translated to a randomly initialized embedding. To the embedded tokens, a (also randomly initialized) positional embedding is added to provide position-relative information. The sequence of tokens is then processed by several architecture dependent Transformer-layers (cf. Section 4). Finally, each contextualized embedding is projected to a single output. A softmax-crossentropy loss is applied over the sequence dimension. See Figure 4 (middle) for a visualization. To make the task input dependent, we generate the data as given in the pseudo code in Figure 4 (left). Note that the `argmin` and `argmax` make this task quite challenging from a learning perspective as the networks start from random embeddings which do not provide any ordering information. Which embeddings correspond to bigger integers and which to smaller

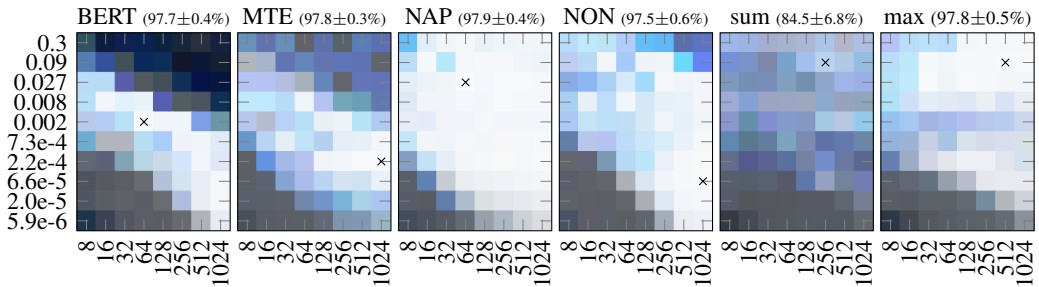

Figure 5: Learning rate (y-axis) vs. model dimension $d$ (x-axis) on the argmin-first-argmax case distinction task (with output across all tokens). The pixels' R (red), G (green) and B (blue) values correspond to min-, mean- and max-accuracy, respectively, of the corresponding hyperparameter combination – see main text for details. The plot shows the validation accuracy when validating on sequences of length $N = 64$. Crosses indicate the combination for best validation accuracy, which we report with standard deviation behind the model name.

integers has to be inferred during training. Further, the case distinction in this task lets us tweak the data bias towards each sub-task. Specifically, we consider a vocabulary size of $S = 100$ integers (0-99) and uniformly random sampled sequences of $N = 128$ tokens in length. This leads to a bias as $p_{argmin} = 1 - (1 - \frac{1}{S})^N \approx 72.4\%$ of data points require the network to pin-point the minimum in the input sequence, $p_{first} \approx 20.1\%$ require to pin-point the first token of the sequence and $p_{argmax} \approx 7.5\%$ require to pin-point the maximum in the input.

### 5.1.1 VARYING MODEL DIMENSION $d$

As a first investigation, we are interested in how varying the model dimension $d$ influences the architectures ability to learn the given task. For this, we train each of the architectures for each of the model dimensions $d \in \{8, 16, 32, 64, 128, 256, 512, 1024\}$ using 10 different learning rates and 5 random seeds for each hyperparameter combination. As we want to base our insights on as many results as possible, we derive a novel, human friendly visualization of results. Figure 5 shows the first results as follows: The outcome of each hyperparameter combination is reported as an RGB pixel in the plot, where the R (red) value corresponds to the accuracy of the worst performing random seed, the G (green) value corresponds to the average over the random seeds and the B (blue) value corresponds to the best performing random seed. For each value (R, G and B), the max over the course of training is taken. This assignment roughly translates as follows: The brighter, the better - brighter pixels correspond to higher min-, mean- and max-accuracy. Blue/turquoise pixels highlight a large performance variation across random seeds and black/grey pixels correspond to hyperparameter combinations where none of the random seeds could solve the task. See Appendix L for figures with color channels split. Given that all architectures are applicable to sequences of any length, we investigate how the architectures generalize to sequences of different length. Specifically, we validated each of the models trained on sequences of length $N = 128$ after every 100 batches on 32 batches with sequences of half the length ($N = 64$). The condensed results in Figure 5 give rise to the following observations: (1) Most models have some hyper-parameter combinations that learn the task well (white pixels). (2) The optimal learning rate depends on the model size, especially in the *BERT* architecture. This has profound implications for hyperparameter optimization: Tuning hyperparameters independent of each other might lead to sub-optimal results. (3) Models with probability simplex limitations (*BERT* and *MTE*) work for a smaller range of hyperparameters. (4) Our *NAP* architecture seems to be the most robust to this generalization. We provide case learning curves and additional results in Appendices E and F. Specifically, to be sure, we repeat this experiment with $L = 4$ and $L = 6$ Transformer layers, but the results stay the same (see Appendix F.1). We also provide the results of an additional model with learned aggregation weights in Appendix H.

### 5.1.2 CASE ACCURACY UNDER VARYING DATA BIASES

As a next experiment we reset the model dimension to $d = 128$ and investigate the models under varying data biases. We do so in two distinct ways: First, we vary the sequence length

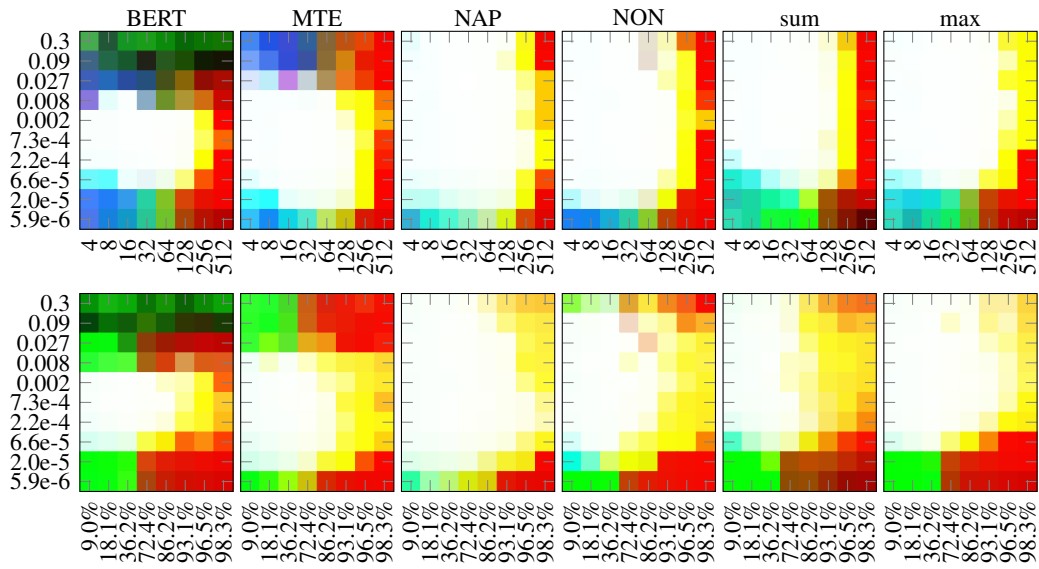

Figure 6: Biased data results on the case distinction task (with output across all tokens). RGB pixel values correspond to *argmin*-, *first*- and *argmax*-mean-case-accuracies, respectively. **Top row:** Learning rate (y-axis) vs. sequence length $N$ (x-axis). **Bottom row:** Learning rate (y-axis) vs. percentage of *argmin*-case in the data (x-axis) with fixed $N = 128$.

$N \in \{4, 8, 16, 32, 64, 128, 256, 512\}$. Note that this implicitly varies the biases $p_{argmin}$, $p_{first}$ and $p_{argmax}$ in the data. Second, we fix the sequence length at $N = 128$ and vary $p_{argmin}$ by explicitly adding or removing corresponding data points. We report the case specific accuracies in Figure 6 as follows: After every 100 batches, we validate the models on 1000 examples per case. Reported is the best accurracy over the course of training in form of pixel value with R (red) corresponding to the *argmin*-case accuracy, G (green) corresponding to the *first*-case accuracy and B (blue) corresponding to the *argmax*-case accuracy. As a consequence, white pixels correspond to all cases learned and yellow pixels correspond to the *argmin*- and *first*-case learned. We make the following observations: (1) If the learning rate is too low, models tend to focus on the majority case (indicated in a shift from blue to red as the bias shifts from the *argmax*- to the *argmin*-case with increasing sequence length $N$ and the shift from green to red for fixed $N$). (2) If the learning rate is too high, the *BERT* architecture tends to focus on the *first*-case. We believe this is due to the architectural bias towards local information as discussed in Section 3. Note that the *first*-case can be solved by relying on the local positional embedding. (3) Only the *NAP* and *max* architecture manage to learn all three cases from highly biased data (see $N = 256$ and $p_{argmin} = 96.5\%$). This shows that the $NAP$ architecture is not only robust to hyperparameters but also robust to biases in the data. In Appendix F.2 and F.3 we provide further experiments investigating different batch sizes and initialization scales. In Appendix F.4 we investigate the local vs. global aggregation focus further.

## 5.2 FIRST TOKEN OUTPUT

The task so far requires the architectures to learn an information flow between tokens to distinguish the case and decide per token, whether it is the token that is looked for or not. Now we investigate, whether all this information can also be aggregated into a single token. We therefore modify the architecture output slightly in that we only take the contextualized embedding of the first token and project from it to a vector of size $N$ (see example on the right in Figure 4). Note that this task set-up is harder and can highlight bottlenecks in the information flow across tokens. We fix the sequence lenght to $N = 128$ and again vary the model dimension $d$. We report the case specific mean accuracies in Figure 7. Min-, mean- and max-overall-accuracies are given in Appendix F.5. We observe: (1) All architectures learn for (almost) all combinations the now close to trivial *first*-case, even though it is not the majority case. (2) The *sum pooling* architecture does not learn any of the other cases. (3) Only *NAP* and *max* learn all three cases in some hyperparameter combinations.

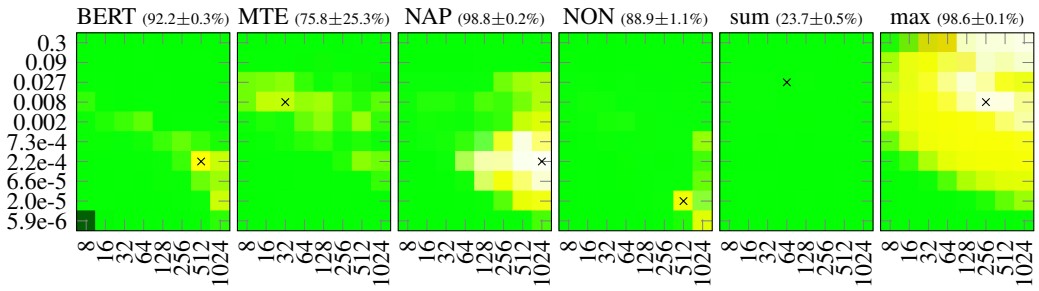

Figure 7: Learning rate (y-axis) vs. model dimension $d$ (x-axis) on the case distinction task with output from the first token. RGB pixel values correspond to the case accuracies. Crosses indicate the best accuracy, reported behind the model name.

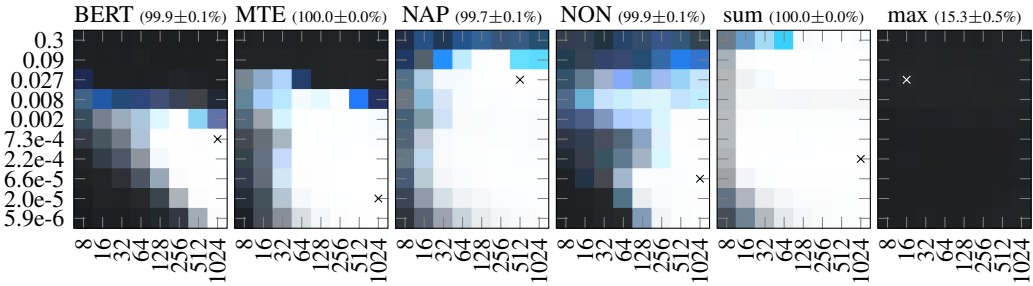

Figure 8: Learning rate (y-axis) vs. model dimension $d$ (x-axis) on the mode finding task. RGB pixel values correspond to min, mean and max accuracy. Crosses indicate the combination for best accuracy, reported behind the model name.

The worse performance of *NON* highlights the advantage of online normalization of the logits. While the softmax provides some form of online normalization, we hypothesize that the worse performance of *MTE* and *BERT* in this task stems from an information bottleneck induced by the probability simplex limitations. To test this hypothesis further, we vary the number of attention heads $M$. The results are shown in Figure 9. We observe that increasing the number of heads helps the *MTE* and *BERT* architecture, supporting our hypothesis. Note however, that *MTE* and *BERT* are still outperformed significantly by *NAP*. In Appendix F.6 we provide a further experiment, varying the depth up to $L = 64$. Results are complementary.

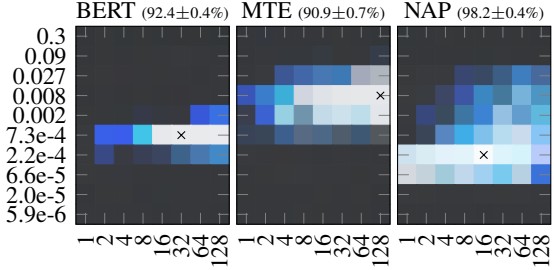

Figure 9: Learning rate (y-axis) vs. attention heads $M$ (x-axis) on the case distinction task (first token output). RGB pixel values correspond to min, mean and max accuracy. Crosses indicate the combination for best accuracy, reported behind the model name.

## 5.3 MODE FINDING TASK

Given the results so far, one could conclude that *max* is the best choice due to its simplicity. Note however, that *max* has an architectural prior that is in line with the underlying task of finding the maximum or minimum of the sequence. To study the effect of architectural priors, we experiment on an additional task: Finding the mode/most common integer in the input sequence. Also this task has ties to NLP (e.g., sentiment analysis), graph neural networks (e.g., consensus/agreement) and reinforcement learning (e.g., count based exploration). Here we remove the positional embeddings, as this task can also be done on sets, and project from the contextualized embedding of the first token to a vector of dimension $S$ (the vocabulary size) over which we apply the softmax-cross-

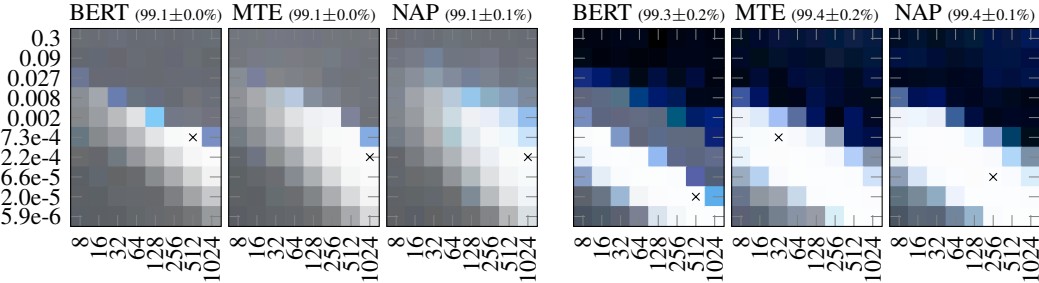

Figure 10: Learning rate (y-axis) vs. model dimension $d$ (x-axis). RGB pixel values correspond to min, mean and max validation performance. Crosses indicate the best combination (validation), from which we report the test performance. For *NON*, *sum* and *max* see Appendices I and J. **Left:** Protein-protein-interaction task. Shown is the node classification F1-score. **Right:** Altered working memory graph agent in the Baby-AI level 3 reinforcement learning task. Shown is the success rate.

entropy loss. We keep $N = 128$ but reduce $S$ to 10 to have meaningful modes. Ties are broken by taking the smallest integer of the ones with maximal occurrence. Results of varying the model dimension $d$ are reported in Figure 8. We observe: (1) *sum pooling* works well on this task, as it has a suitable architecutral prior. (2) *max pooling* cannot learn the task, not even with a model dimension $d = 1024 = 8 \cdot N$. In Appendix G.1 we provide complementary results varying the vocabulary size. We refer an interested reader to Xu et al. (2020) for more on architecture-task alignment.

## 5.4 Generalization to Graph Neural Networks and Reinforcement Learning

NLP experiments normally require large amounts of data and large models – a setting which would make an investigation of the training dynamics under various hyperparameters prohibitively expensive. We therefore focus in our generalization study on the two other domains where attention based architectures have recently gained traction. Specifically, to see whether the the learning dynamics seen so far generalize, we perform two additional experiments: One investigating the architectures in a graph neural network (GNN) task and one investigating them in a reinforcement learning (RL) setup. Specifically, we train a transformer based GNN in the Protein-Protein-Interaction (PPI) graph node classification task (Zitnik & Leskovec, 2017) and investigate the different architectures as alterations on the working memory graph architecture for RL (Loynd et al., 2020), training on level 3 of the Baby-AI environment (Chevalier-Boisvert et al., 2019). The validation results under varying model dimension $d$ are given in Figure 10. The results replicate several observations made so far. Specifically: (1) All attention based architectures can learn the task for some hyperparameter combinations. (2) There is a correlation between optimal learning rate and model dimension. (3) *BERT* is hyperparameter sensitive in comparison. See Appendices I and J for experiment details.

## 6 Conclusion

Taking all observations together, we conclude: Many recent works apply some sort of self-attention mechanism involving a softmax that projects the attention weights to the probability simplex. In this work we question the softmax in dot-product self-attention modules. Our theoretical investigation shows that softmax-attention outputs are constrained to the convex hull spanned by the value vectors. In our experiments we show that this can lead to an unwanted hyperparameter sensibility. We show that simpler architectures like max- and sum-pooling perform well when their architectural prior aligns with the underlying task. These architectures however fail in cases where the architectural prior is not suitable. As a solution, we propose to replace the softmax in attention through normalization. Our resulting normalized attention pooling (*NAP*) architecture is the only architecture of the 6 investigated that is robust in all tasks and setups, even if the data is heavily skewed towards easier sub-tasks. We hope that our work provides a stepping stone to examine architectures under varying biases in the data. Further, we see potential in exploring the correlated effects of hyperparameters. Please also refer to our broader impact statement in Appendix K.

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

## A    REPRESENTATION POWER WITH RESPECT TO XOR

**Lemma 1.** *No convex combination can represent the binary exclusive OR (XOR) function defined on binary inputs $x_1 \in \{0, 1\}$ and $x_2 \in \{0, 1\}$ by the indicator function as $XOR(x_1, x_2) = \mathbf{1}_{x_1 \neq x_2}$.*

*Proof.* Suppose there exist convex combination weights $a_1$ and $a_2$ with $a_1 + a_2 = 1$, such that $a_1 \cdot x_1 + a_2 \cdot x_2$ represents the XOR function. Plugging in $x_1 = x_2 = 1$ yields $a_1 \cdot x_1 + a_2 \cdot x_2 = a_1 + a_2 = 1$, which gives the contradiction. $\square$

**Lemma 2.** *Given the two binary inputs $x_1 \in \{0, 1\}$ and $x_2 \in \{0, 1\}$, there exists an affine mapping $f : \{0, 1\}^2 \to \mathbb{R}^2$, such that the normalized weighting given by*

$$\frac{f_1(x_1, x_2) - \mu_{f(x_1,x_2)}}{\sigma_{f(x_1,x_2)}} \cdot x_1 + \frac{f_2(x_1, x_2) - \mu_{f(x_1,x_2)}}{\sigma_{f(x_1,x_2)}} \cdot x_2$$

*is equivalent to the logical exclusive OR given by the indicator function as $XOR(x_1, x_2) = \mathbf{1}_{x_1 \neq x_2}$.*

*Proof.* For a vector $\boldsymbol{l} \in \mathbb{R}^2$, the standard deviation $\sigma_{\boldsymbol{l}}$ can be simplified to

$$\sigma_{\boldsymbol{l}} = \sqrt{\frac{1}{2} \sum_{i \in \{1,2\}} (l_i - \mu_{\boldsymbol{l}})^2}$$

$$= \sqrt{\frac{1}{2} \left( \left( l_1 - \frac{l_1 + l_2}{2} \right)^2 + \left( l_2 - \frac{l_1 + l_2}{2} \right)^2 \right)}$$

$$= \frac{1}{2} |l_1 - l_2|$$

and the normalization function reduces to

$$normalize(\boldsymbol{l}) = \left[ \frac{l_1 - \mu_{\boldsymbol{l}}}{\sigma_{\boldsymbol{l}}}, \frac{l_2 - \mu_{\boldsymbol{l}}}{\sigma_{\boldsymbol{l}}} \right]^T$$

$$= \left[ \frac{l_1 - l_2}{|l_1 - l_2|}, \frac{l_2 - l_1}{|l_1 - l_2|} \right]^T$$

$$= \begin{cases} [1, -1]^T & if \quad l_1 > l_2 \\ [-1, 1]^T & if \quad l_1 < l_2 \\ undef. & if \quad l_1 = l_2 \end{cases}$$

As an example, consider the affine mapping $f(\boldsymbol{x}) = \boldsymbol{l} = [3x_1 + 1, 2x_2]^T$, which for $x_1 \in \{0, 1\}$ and $x_2 \in \{0, 1\}$ results in the function

$$\frac{3x_1 + 1 - 2x_2}{|3x_1 + 1 - 2x_2|} \cdot x_1 + \frac{-3x_1 - 1 + 2x_2}{|3x_1 + 1 - 2x_2|} \cdot x_2 = \mathbf{1}_{x_1 \neq x_2}$$

$\square$

We note that for a realization of such an affine mapping across tokens given the weight sharing constraints of the discussed architectures we would need $x_1$ and $x_2$ to be distinguishable for the mapping to keys and queries, e.g., through positional embeddings. This however does not invalidate our conclusion that normalized weighting is more expressive than softmax weighting, as we do not require the inputs that are weighted to be distinguishable.

## B    VANISHING GRADIENT ANALYSIS

For better readability we will use subscripts here to index sequence elements and omit the implicit reference to the head and vector dimension used in the main text.

**Lemma 3.** *The partial derivative $g = \frac{\delta \mathrm{softmax}(\boldsymbol{l})_i}{\delta l_i}$ is limited to the range $g \in (0, 0.25]$.*

*Proof.*

$$g = \frac{\delta \text{softmax}(\boldsymbol{l})_i}{\delta l_i} = \frac{\exp{(l_i)} \sum_j \exp{(l_j)} - \exp{(2 \cdot l_i)}}{\left(\sum_j \exp{(l_j)}\right)^2} = a_i \cdot (1 - a_i) \text{ with } a_i = \text{softmax}(\boldsymbol{l})_i$$

Maximizing this function we get that the maximum of $g = 0.25$ is achieved for $a_i = 0.5$. Noting that $a_i > 0$, the claim follows. □

To expand this further, we introduce the notion of $\alpha$-bandwidth.

**Definition 1.** *We call the range $\Delta x$ in which the absolute value of a gradient factor $\frac{\delta f(x)}{\delta x}$ exceeds a threshold $\alpha$, i.e. $\left|\frac{\delta f(x)}{\delta x}\right| > \alpha$, its $\alpha$-bandwidth $BW_\alpha \left(\frac{\delta f(x)}{\delta x}\right)$.*

Please refer to Figure 11 for a visualization of this definition. Intuitively, the $\alpha$-bandwidth gives a measure for the range of operation, where at least an $\alpha$-fraction of the gradient gets backpropagated.

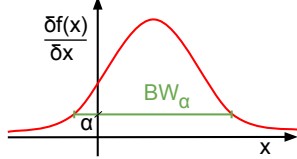

It also provides a measure of how fast a function saturates: Given $x$ with $\frac{\delta f(x)}{\delta x} > \alpha$, updating $x$ to $x' = x \pm BW_\alpha \left(\frac{\delta f(x)}{\delta x}\right)$ puts $x'$ into the saturated region where $\frac{\delta f(x')}{\delta x'} \leq \alpha$. A function with a small $\alpha$-bandwidth can therefore more easily saturate than a function with a larger $\alpha$-bandwidth.

Figure 11: Visual definition

**Lemma 4.** *The $\alpha$-bandwidth of $g = \frac{\delta \text{softmax}(\boldsymbol{l})_i}{\delta l_i}$ can be calculated as*

$$BW_\alpha(g) = \ln \left( \frac{1 - 2\alpha + \sqrt{1 - 4\alpha}}{1 - 2\alpha - \sqrt{1 - 4\alpha}} \right)$$

*Proof.* Substituting $x = \exp(l_i)$ and $y = \sum_{j \neq i} \exp(l_j)$ we can write $g$ as $g(x) = \frac{x}{x+y} \cdot \left(1 - \frac{x}{x+y}\right)$.

Solving $g(x) = \alpha$ yields $x = \frac{y}{2\alpha} \left(1 - 2\alpha \pm \sqrt{1 - 4\alpha}\right)$. Reversing the substitution of $l_i = \ln(x)$ and subtracting the larger from the smaller value, we get:

$$BW_\alpha(g) = \ln \left( \frac{y}{2\alpha} \left(1 - 2\alpha + \sqrt{1 - 4\alpha}\right) \right) - \ln \left( \frac{y}{2\alpha} \left(1 - 2\alpha - \sqrt{1 - 4\alpha}\right) \right)$$

$$= \ln \left( \frac{1 - 2\alpha + \sqrt{1 - 4\alpha}}{1 - 2\alpha - \sqrt{1 - 4\alpha}} \right)$$

□

**Corollary 1.** *The range in which $g = \frac{\delta \text{softmax}(\boldsymbol{l})_i}{\delta l_i} > 0.01$ is smaller than 9.2.*

*Proof.* Calculating the $\alpha$-bandwidth for $\alpha = 0.01$ we get $BW_{0.01}(g) \approx 9.170$. □

Note that $BW_\alpha(g)$ is independent of the length of the vector $\boldsymbol{l}$ as the remainder $y = \sum_{j \neq i} \exp(l_j)$ cancels. For the self-attention in the Transformer this translates to the fact that $BW_\alpha(g)$ is independent of the sequence length. This means that the Transformer architecture cannot adapt this range of operation according to the sequence length. Aggregated information from long sequences inevitably needs to be averaged or scaled to fall within this range.

We further note that the constant $0.01$-bandwidth on its own is not the problem. In fact, a tanh-non-linearity has a smaller $0.01$-bandwidth. However, in the standard Transformer architecture, gradients are scaled dependent on the layer input. More specifically, dropping sequence indices to simplify notation, we have

**Lemma 5.** *For $\boldsymbol{k} = W_k \boldsymbol{x} + \boldsymbol{b}_k$, $\boldsymbol{q} = W_q \boldsymbol{x} + \boldsymbol{b}_q$, $W_q^T W_k + W_k^T W_q$ invertible and $||\boldsymbol{x}|| \geq c_1$ we can find constants $c_2 > 0$ and $c_3 > 0$ such that*

$$\left\| \frac{\delta l}{\delta \boldsymbol{x}} \right\| \geq c_2 \cdot ||\boldsymbol{x}|| - c_3$$

*Proof.* Writing out $l = \frac{<q,k>}{\sqrt{d_h}}$ we get

$$l = \frac{1}{\sqrt{d_h}}(\boldsymbol{x}^T W_q^T W_k \boldsymbol{x} + (\boldsymbol{b}_k^T W_q + \boldsymbol{b}_q^T W_k)\boldsymbol{x} + \boldsymbol{b}_q^T \boldsymbol{b}_k)$$

Using $\frac{\delta \boldsymbol{x}^T A \boldsymbol{x}}{\delta \boldsymbol{x}} = \boldsymbol{x}^T(A + A^T)$ and substituting $\boldsymbol{g_b} = \boldsymbol{b}_k^T W_q + \boldsymbol{b}_q^T W_k$ we get

$$\frac{\delta l}{\delta \boldsymbol{x}} = \frac{1}{d_h}(\boldsymbol{x}^T(W_q^T W_k + W_k^T W_q) + \boldsymbol{g_b})$$

Setting $\boldsymbol{g_x} = \boldsymbol{x}^T(W_q^T W_k + W_k^T W_q)$ and solving for $\boldsymbol{x}$ we get $(W_q^T W_k + W_k^T W_q)^{-1}\boldsymbol{g_x}^T = \boldsymbol{x}$. With this, we can bound $||\boldsymbol{g_x}||$ with the operator norm on $(W_q^T W_k + W_k^T W_q)^{-1}$. Specifically, if we set $c_2$ as

$$c_2 = \frac{1}{\sqrt{d_h} \cdot ||(W_q^T W_k + W_k^T W_q)^{-1}||}$$

we get $\frac{1}{\sqrt{d_h}} \cdot ||\boldsymbol{g_x}|| \geq c_2 \cdot ||\boldsymbol{x}||$. We can then bound $||\frac{\delta l}{\delta \boldsymbol{x}}||$ as

$$\left\|\frac{\delta l}{\delta \boldsymbol{x}}\right\| = \frac{1}{\sqrt{d_h}}||\boldsymbol{g_x} + \boldsymbol{g_b}|| \geq \frac{1}{\sqrt{d_h}}\left|||\boldsymbol{g_x}|| - ||\boldsymbol{g_b}||\right|$$

Therefore, if we set $c_1 = \frac{1}{c_2 \cdot \sqrt{d_h}} \cdot ||\boldsymbol{g_b}||$ we get $||\boldsymbol{g_x}|| \geq c_2 \cdot \sqrt{d_h} \cdot ||\boldsymbol{x}|| \geq ||\boldsymbol{g_b}||$ and hence

$$\left\|\frac{\delta l}{\delta \boldsymbol{x}}\right\| \geq \frac{1}{\sqrt{d_h}}(||\boldsymbol{g_x}|| - ||\boldsymbol{g_b}||) \geq c_2 \cdot ||\boldsymbol{x}|| - c_3$$

for $c_3 = \frac{1}{\sqrt{d_h}} \cdot ||\boldsymbol{g_b}||$. $\qquad\square$

Lemma 5 shows that the magnitude of the gradient factor scales with the magnitude of the input. Therefore, a single large input can yield a correspondingly large parameter update. This in turn can put the softmax easily into the saturated region, since the softmax only has a constant $\alpha$-bandwidth. Once saturated, the softmax leads to vanishing gradients henceforth.

## C  SEQUENCE LENGTH DEPENDENT FOCUS

For Figure 2 we sample 16'384 value, key and query vectors of dimension $d_h = 128$ per sequence length $N \in \{1, 2, 4, 8, 16, 32, 64, 128, 512, 1024, 2048\}$ from a normal Gaussian $\mathcal{N}(\boldsymbol{0}, \boldsymbol{I}_{d_h})$ - $\boldsymbol{I}_{d_h}$ being the $d_h$-dimensional identity matrix. We split the samples to form the sequences and calculate the corresponding output vectors $\boldsymbol{o}^i$ for $i \in \{1, \ldots, N\}$. Here, the softmax attention outputs are calculated as described in Section 2, while the mean-, sum- and max-outputs are calculated as mean-, sum- and max-reduce of the value vectors over the sequence dimension. For the normalized results we take the sum-output vectors and normalize them (over the $d_h$-dimensional vector dimension). Note that such a normalization can be applied to any of the aggregation methods to get qualitatively similar results. The plots in Figure 2 are generated by reporting the standard deviation over all output values and the mean norm of the output values, respectively.

Given the numerous successes of Transformers in natural language processing, we conjecture that a bias towards local information might be beneficial in language modeling. However, the implicit dependence on sequence length in a model that should be oblivious to different input sequence lengths is questionable. We leave an in depth investigation to future work.

## D  ARCHITECTURES

We provide a schematic of 1 Transformer-layer of each architecture investigated in Figure 12. Our base architectures consist of 2 such layers followed by a projection to the output dependent on the task as described in the corresponding sections (cf. Section 5.1, 5.2 and 5.3).

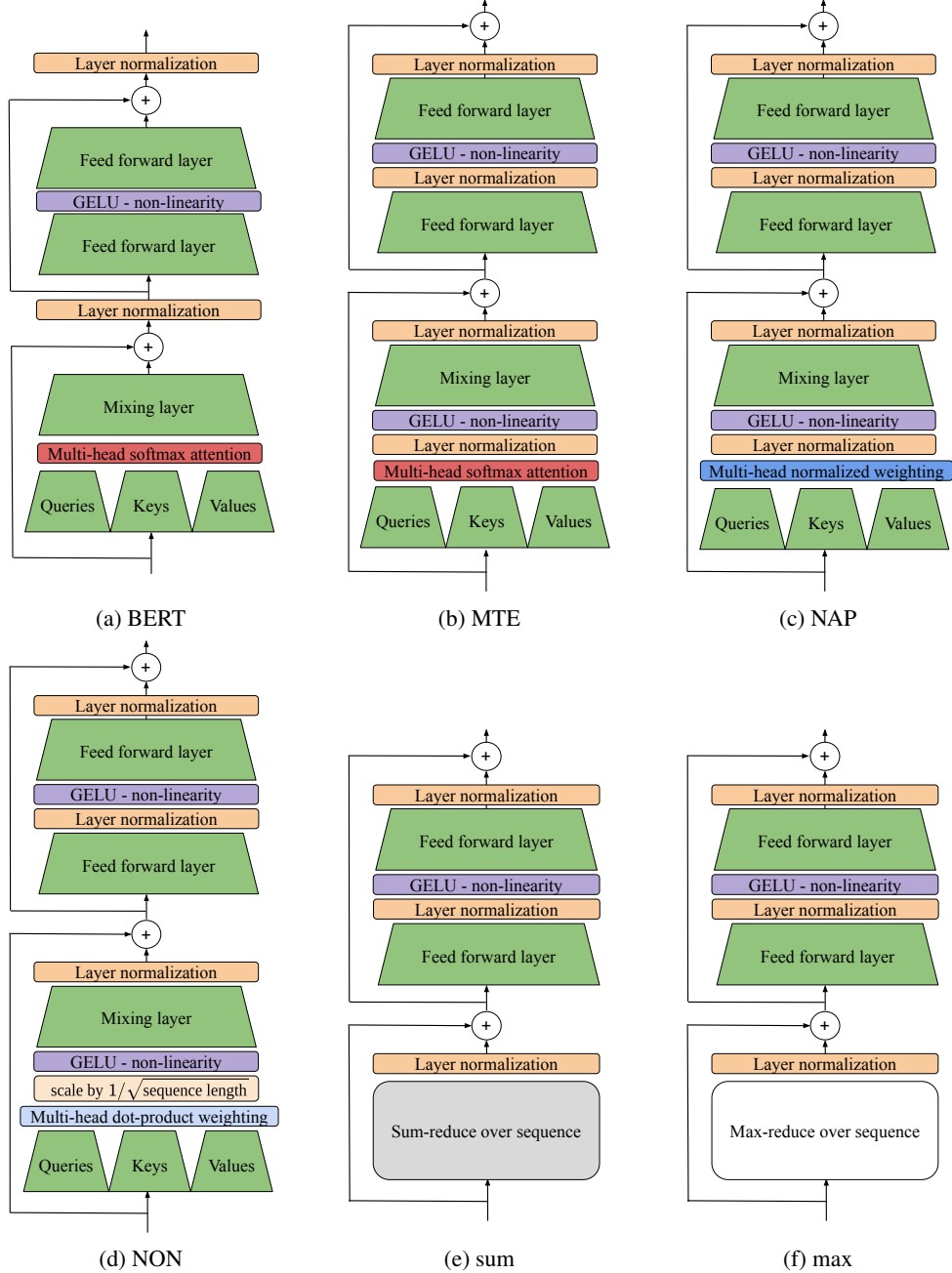

Figure 12: Schematics of 1 Transformer-layer block of the different architectures investigated. Green layers correspond to the main weight matrices that are trained. Note that displayed dimensions are not to scale - the hidden dimension of the feed forward layer is larger than the model dimension and the hidden layer size in the feed forward network of "max" and "sum" are adjusted to approximately match the parameter count of the other architectures.

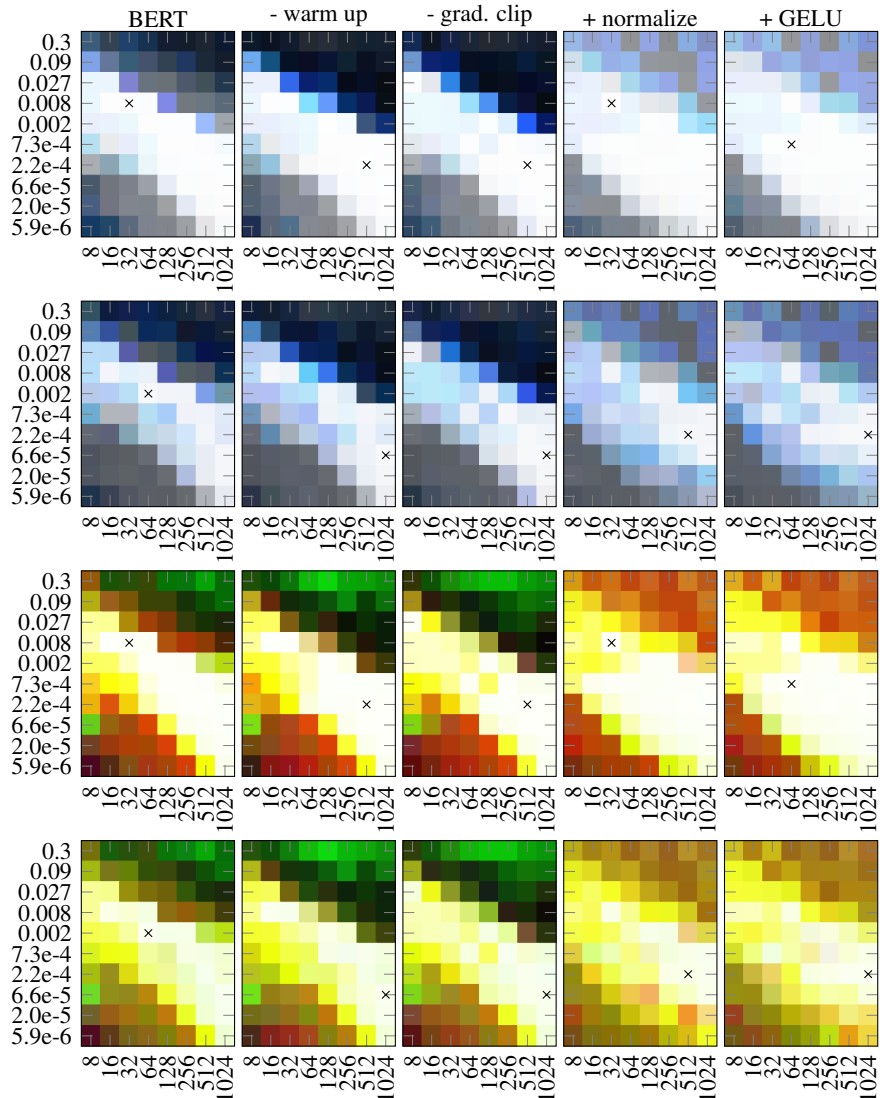

Figure 13: Learning rate (y-axis) vs. model dimension $d$ (x-axis) on the argmin-first-argmax case distinction task (with output across all tokens) - architecture modification ablation study. In the first two rows, RGB pixel values correspond to min-, mean- and max-accuracy. In the last two rows, RGB pixel values correspond to *argmin*-, *first*- and *argmax*-mean-case-accuracies. **1. row:** Training accuracy (sequence length $N = 128$). **2. row:** Validation accuracy when validating on sequences of half the length ($N = 64$). **3. row:** Training case accuracy (sequence length $N = 128$). **4. row:** Validation case accuracy when validating on sequences of half the length ($N = 64$). Crosses indicate the combination for best mean accuracy, which are reported in Table 1.

## D.1 ARCHITECTURE MODIFICATION ABLATIONS

An empirical ablation of the modifications that lead from the *BERT* architecture to the *MTE* architecture is given in Figure 13. The plots are generated as described in Sections 5.1.1 and 5.1.2.

The first column in Figure 13 corresponds to the original *BERT* architecture, trained with gradient norm clipping and learning rate warm up.

The second column (- warm up) corresponds to the same architecture, but trained without learning rate warm up. Here we see that too high learning rates learn even less without learning rate

Table 1: Ablation study accuracy values taken from the hyper-parameter combination that led to the best mean overall accuracy, indicated by a cross in Figure 13.

| | | BERT | - warm up | - grad. clip | + normalize | + GELU |
|---|---|---|---|---|---|---|
| Overall | min | 99.3% | 99.4% | 99.3% | 99.2% | 99.3% |
| Training | mean | 99.4% | 99.5% | 99.4% | 99.3% | 99.3% |
| Accuracy | max | 99.5% | 99.6% | 99.6% | 99.5% | 99.5% |
| Overall | min | 96.5% | 96.9% | 96.9% | 96.3% | 96.7% |
| Validation | mean | 97.2% | 97.6% | 97.2% | 96.8% | 97.3% |
| Accuracy | max | 98.2% | 98.2% | 98.2% | 98.4% | 98.2% |
| Mean Case | argmin | 99.3% | 99.6% | 99.5% | 99.6% | 99.5% |
| Accuracy | first | 100% | 100% | 100% | 100% | 100% |
| Training | argmax | 96.9% | 98.1% | 97.5% | 96.5% | 98.0% |
| Mean Case | argmin | 98.0% | 98.0% | 98.0% | 98.1% | 97.9% |
| Accuracy | first | 100% | 100% | 100% | 100% | 100% |
| Validation | argmax | 93.9% | 93.8% | 93.1% | 91.2% | 93.8% |

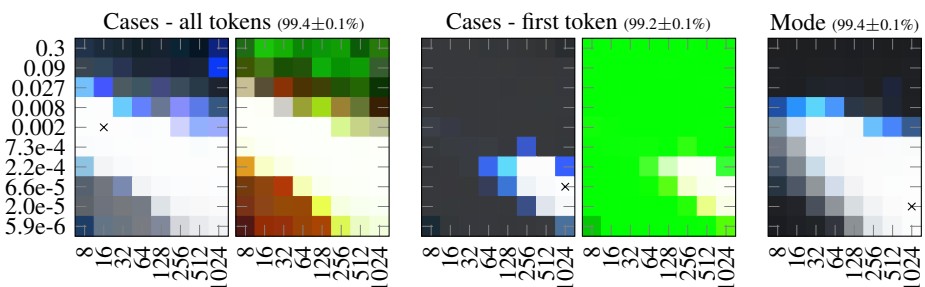

Figure 14: Learning rate (y-axis) vs. model dimension $d$ (x-axis) in the different task setups when just replacing the softmax in *BERT* with normalization. The plots from left to right are grouped according to the tasks: The case task with outputs across all tokens (cf. Section 5.1), the case task with outputs from the first token (cf. Section 5.2) and the mode task (cf. Section 5.3). For the case tasks, the left sub-plot reports min, mean and max accuracies while the right sub-plot reports mean case accuracies as RGB pixel values. The mode plot reports min, mean and max accuracies. The plots show the accuracies for $N = 128$.

warm up in the *BERT* architecture, hinting at a necessity for learning rate warm up for the original architecture.

The third column (- grad. clip) reports the results if we further remove gradient clipping from the training schedule. This does not seem to have a big impact in our setup.

Next, we report in the forth column (+ normalize) the results of moving the layer normalization before the residual addition and introducing an additional layer normalization right after the attention mechanism as well as on the hidden layer of the feed forward network. Note that this change removes the bias towards local information discussed in the end of Section 3. We see that this change leads to a profound shift in focus in regions where the learning rate is high: models with the original normalization focus on the (local) *first*-case, while models with our normalization focus on the (majority) *argmin*-case. This is in line with the insights stated in Section 5.1.2.

Finally, we report in the fifth column (+ GELU) the results of adding an additional GELU layer after the attention. These results correspond to the *MTE* architecture used throughout the paper.

Apart from the performance landscape changes just mentioned, the best hyper-parameter accuracies remain similar throughout all modifications, cf. Table 1.

For completeness, we also provide results for just replacing the softmax in *BERT* with normalization in Figure 14, that is, we train NAP with learning rate warmup, gradients clipped, no GELU after the attention and layer normalization as in BERT. The results are complementary to those presented in the main text: the model is still more robust to hyperparameter changes, albeit not performing well

on higher learning rates. On these high learning rates, the model focuses (similar to BERT) on the local first task. This is due to the placement of layer normalization, which with the initialization of the mixing layer yields a focus on local information.

## D.2 REGULARIZATION EXPERIMENTS

To limit the number of variables which are not accounted for in the experiments, we focus on the *infinite data but limited training time* regime. In this regime, every batch consists of new data points. We believe that this regime is of paramount interest in future research, as more devices create a constant stream of data and training is more limited by the available training time than the available data. This regime allows us to omit regularization in all architectures as over-fitting is not an issue. In fact, our supplementary experiments below as well as related work (Lan et al., 2020) show that regularization does not help in this regime. We leave a comparison of the architectures in the limited data regime to future work.

Here, we show empirical results supporting the intuition that $L2$ as well as *dropout* regularization does not help in our setup. For each of our tasks, we take our default hyper-parameters ($d = 128$, $L = 2$, $M = 4$, $N = 128$) and train 5 random seeds per learning rate for models with regularization, varying the dropout rate in $\{0.0625, 0.125, 0.25, 0.5\}$ and the $L2$ regularization weighting in $\{0.0001, 0.001, 0.01, 0.1\}$. Tables 2, 3 and 4 report the best mean accuracy achieved with the small number behind the accuracies indicating the regularization used, 1 referring to the smallest, 4 to the largest. We underline the results where regularization did lead to an improvement in mean accuracy. Note however that these improvements should be taken with a grain of salt, as (1) none of these improvements is significant considering the performance variation across random seeds and (2) the regularized values are likely to be overestimated, as the max is taken over 40 averages (4 regularization values times 10 learning rates) as compared to 10 averages (10 learning rates) in the unregulated case.

Overall we note that none of the architectures consistently benefits from regularization in our setup and regularization often decreases mean performance. Further, we point out that the best performance with regularization is most of the times achieved with the smallest regularization.

Table 2: Regularization results in the case distinction task with output taken across all tokens. The top three rows correspond to the best mean training accuracy, while the bottom three rows correspond to the best mean validation accuracy when validating on sequences of half the length.

|  | BERT | MTE | NAP | NON | sum | max |
|---|---|---|---|---|---|---|
| unregularized | 99.3% | 99.1% | 99.3% | 99.1% | 98.9% | 99.2% |
| with dropout | 98.1%[1] | 97.3%[1] | 97.8%[1] | 97.5%[1] | 97.3%[1] | 98.2%[1] |
| with $L2$-regularization | 99.3%[2] | 99.2%[1] | 99.2%[2] | 99.2%[1] | 98.9%[1] | 99.4%[2] |
| unregularized | 95.5% | 95.5% | 97.0% | 95.3% | 75.0% | 97.1% |
| with dropout | 94.4%[1] | 94.6%[1] | 96.8%[2] | 96.0%[1] | 83.1%[1] | 96.3%[1] |
| with $L2$-regularization | 97.2%[2] | 93.6%[2] | 97.1%[1] | 96.1%[2] | 67.7%[2] | 97.2%[2] |

Table 3: Regularization results in the case distinction task with output from the first token. The top three rows correspond to the best mean training accuracy, while the bottom three rows correspond to the best mean validation accuracy when validating on sequences of half the length.

|  | BERT | MTE | NAP | NON | sum | max |
|---|---|---|---|---|---|---|
| unregularized | 36.6% | 66.5% | 94.5% | 23.2% | 22.8% | 97.8% |
| with dropout | 44.9%[1] | 44.3%[1] | 85.0%[1] | 23.2%[1] | 22.6%[1] | 92.6%[1] |
| with $L2$-regularization | 36.0%[2] | 55.3%[1] | 93.8%[2] | 22.8%[1] | 22.8%[1] | 95.4%[1] |
| unregularized | 36.7% | 50.6% | 83.9% | 29.6% | 28.5% | 88.5% |
| with dropout | 41.4%[2] | 40.7%[1] | 74.6%[1] | 29.6%[3] | 28.9%[4] | 87.8%[1] |
| with $L2$-regularization | 37.2%[2] | 45.7%[1] | 82.5%[2] | 28.9%[1] | 29.0%[1] | 81.0%[1] |

Table 4: Regularization results in the mode finding task. The top three rows correspond to the best mean training accuracy, while the bottom rows correspond to the best mean validation accuracy when validating on sequences of twice the length.

|  | BERT | MTE | NAP | NON | sum | max |
|---|---|---|---|---|---|---|
| unregularized | 99.6% | 99.8% | 99.6% | 98.7% | 99.8% | 14.4% |
| with dropout | 93.9%[1] | 93.3%[1] | 94.3%[1] | 91.8%[1] | 93.3%[1] | 24.5%[1] |
| with $L2$-regularization | 99.5%[1] | 99.9%[2] | 99.7%[1] | 98.8%[1] | 99.9%[4] | 14.4%[2] |
| unregularized | 95.3% | 95.4% | 94.9% | 91.3% | 95.8% | 13.5% |
| with dropout | 94.8%[2] | 95.4%[2] | 93.8%[1] | 92.6%[1] | 95.7%[2] | 13.4%[4] |
| with $L2$-regularization | 94.7%[1] | 96.0%[1] | 94.9%[1] | 94.7%[2] | 95.8%[1] | 13.7%[1] |

# E    CASE LEARNING CURVES

Figures 15, 16 and 17 show the case accuracies over the course of training. The corresponding color plot is given in Figure 18. Besides the observations made in the main text, a few additional insights can be noted: (1) Cases are mostly learned in the order of their occurrences (recall that $72.37\%$ of the examples are from the *argmin* case, $20.09\%$ are from the *first* case and $7.53\%$ are from the *argmax* case). This is to be expected when training with gradient descent, cf. Chatterjee (2020). (2) This order is not always given in the *BERT* architecture.

Besides the focus on the *first* case if the learning rate is too high - discussed in the main text - we also highlight a curiosity that occurs when the model dimension is too small (see plot highlighted in with red in Figure 15): The *first* case is learned and then unlearned in favor of the *argmin* case. Note that all 5 random seeds follow this pattern. Note also that for a different learning rate, the opposite holds as seen in the plot just below the highlighted plot. We highly encourage an interested reader to check out our code release,[3] which includes these results as well as visualization scripts to inspect them further.

---

[3]See supplementary material.

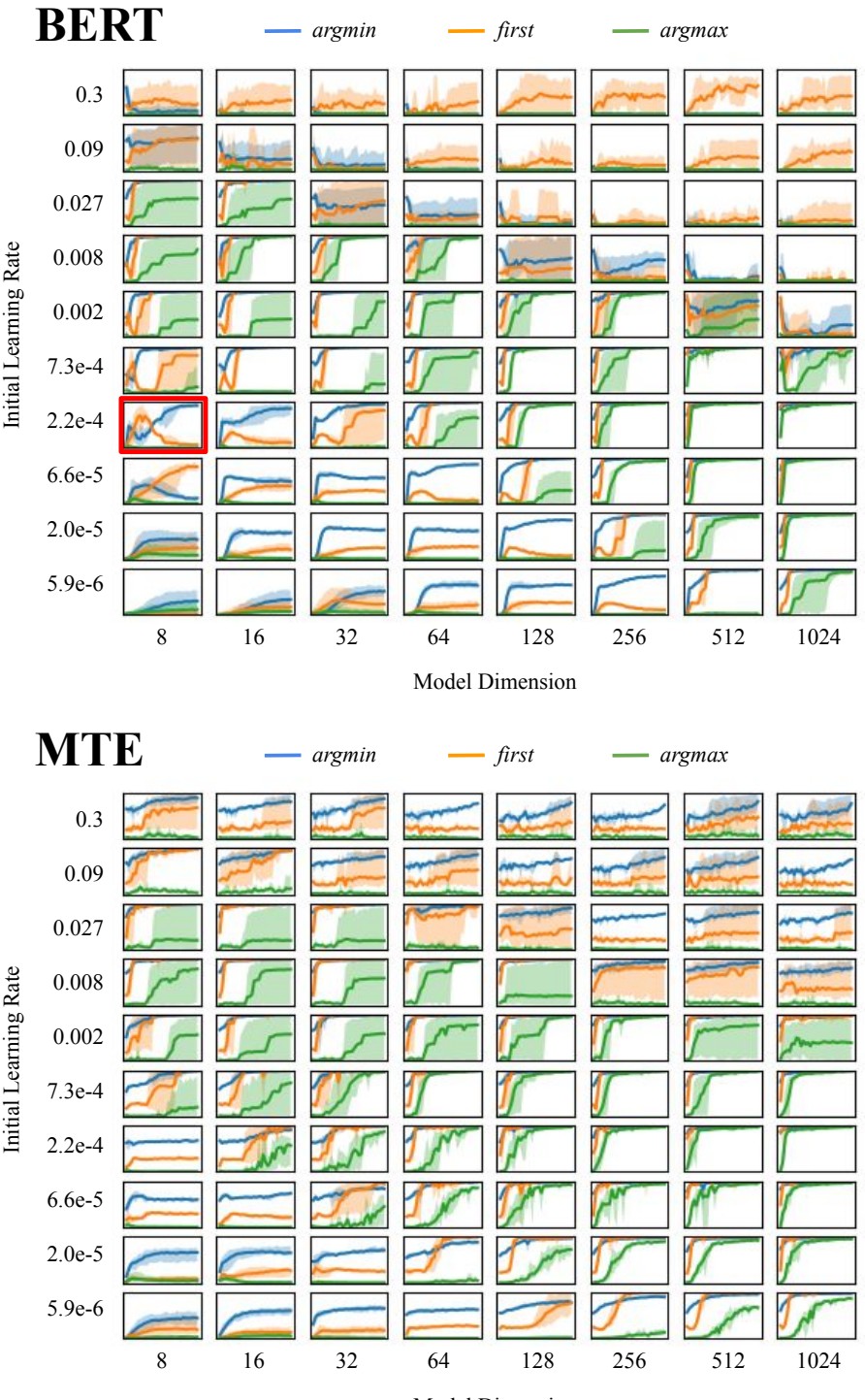

Figure 15: Case accuracies over the course of training on the *argmin-first-argmax* case distinction task with output across all tokens, cf. Section 5.1. Each small sub-plot shows the case accuracies (y-axis, bottom is set to 0%, top to 100%) over the course of training (x-axis). Solid lines represent the mean accuracy over the 5 random seeds while shaded areas fill the spread between min- and max-accuracy achieved. Models *BERT* and *MTE* are shown here, cf. Figures 16 and 17.

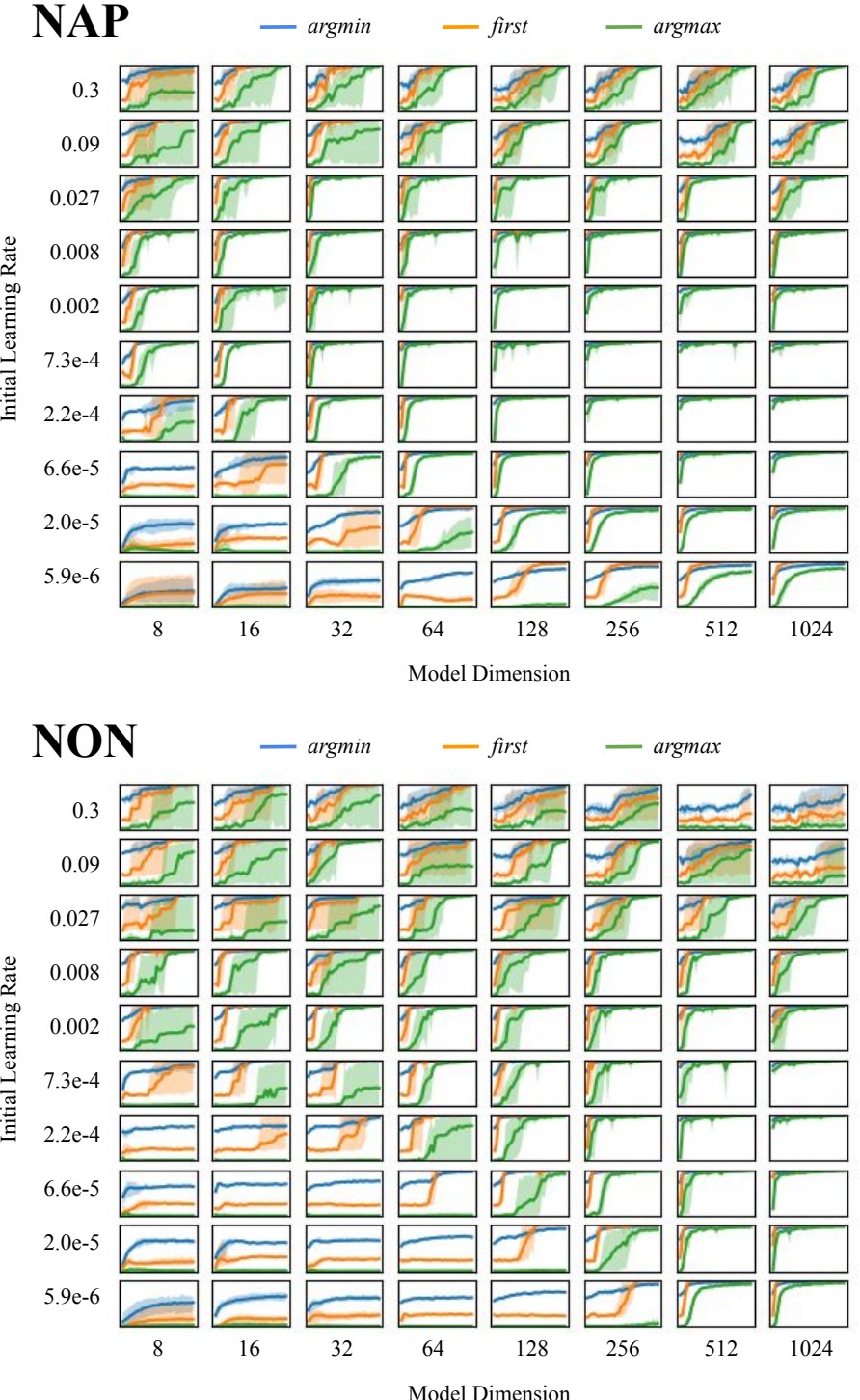

Figure 16: Case accuracies over the course of training on the *argmin-first-argmax* case distinction task with output across all tokens, cf. Section 5.1. Each small sub-plot shows the case accuracies (y-axis, bottom is set to 0%, top to 100%) over the course of training (x-axis). Solid lines represent the mean accuracy over the 5 random seeds while shaded areas fill the spread between min- and max-accuracy achieved. Models *NAP* and *NON* are shown here, cf. Figures 15 and 17.

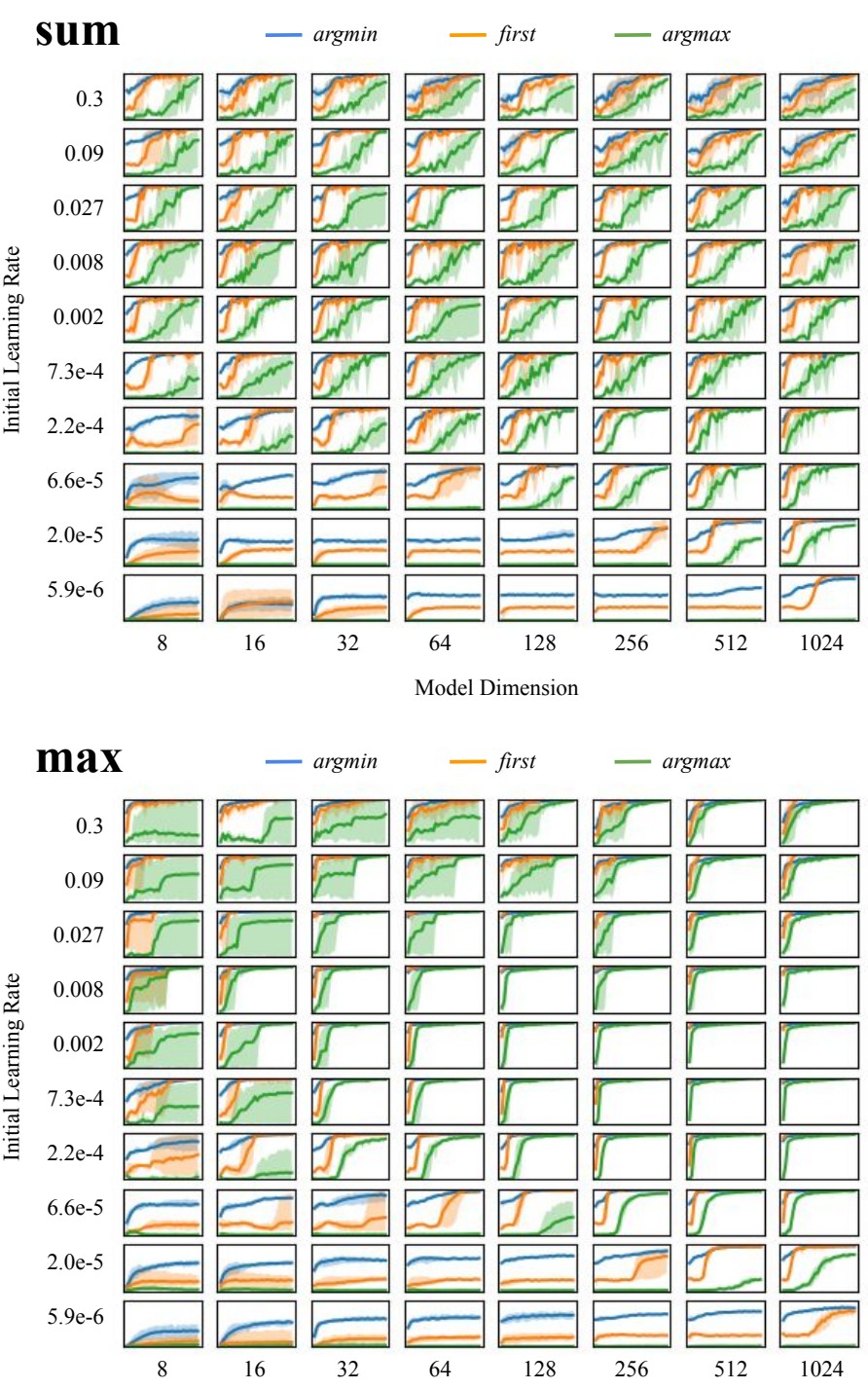

Figure 17: Case accuracies over the course of training on the *argmin-first-argmax* case distinction task with output across all tokens, cf. Section 5.1. Each small sub-plot shows the case accuracies (y-axis, bottom is set to 0%, top to 100%) over the course of training (x-axis). Solid lines represent the mean accuracy over the 5 random seeds while shaded areas fill the spread between min- and max-accuracy achieved. Models *sum* and *max* are shown here, cf. Figures 15 and 16.

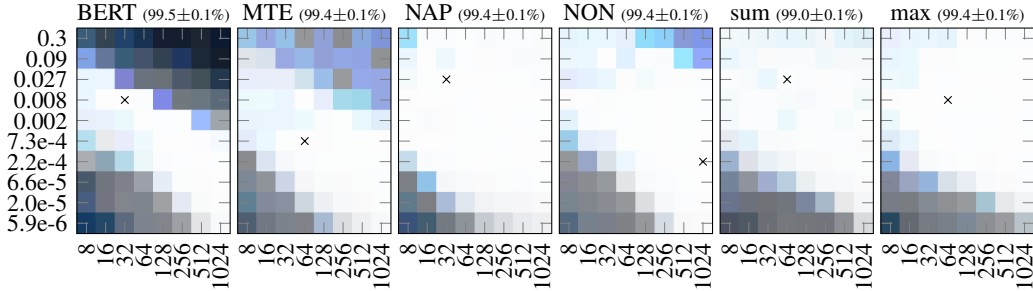

Figure 18: Learning rate (y-axis) vs. model dimension $d$ (x-axis) on the argmin-first-argmax case distinction task (with output across all tokens). The pixels' R (red), G (green) and B (blue) values correspond to min-, mean- and max-accuracy, respectively, of the corresponding hyperparameter combination. The plot shows the accuracy when evaluating on sequences of length $N = 128$. Crosses indicate the combination for best mean accuracy, reported behind the model name.

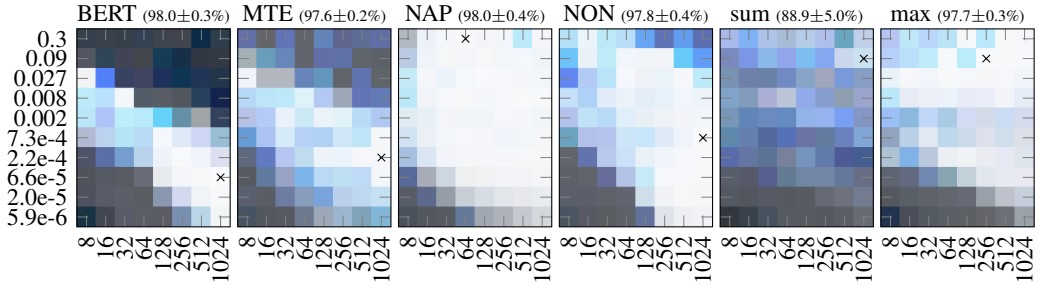

Figure 19: Rerun of the experiments of Figure 5 with $L = 4$ instead of $L = 2$. Crosses indicate the combination for best validation accuracy, which we report behind the model name.

## F  CASE DISTINCTION TASK - ADDITIONAL RESULTS

Figure 18 shows the accuracy in the case distinction task when evaluating with the same sequence length as used during training, i.e., $N = 128$.

### F.1  VARYING MODEL DIMENSION WITH 4 AND 6 LAYERS

Figures 19 and 20 show the results when varying the model dimension for deeper models, specifically 4 and 6 Transformer-layer deep models. The results do not differ which is why we choose to set the default to $L = 2$ layers, which in turn allows us to run more experiments. Note that we also present an experiment in F.6 where we explicitly vary the depth up to $L = 64$ Transformer-layers.

### F.2  VARYING BATCH SIZE

In Figure 21 we provide the case accuracy results of an additional experiment, varying the batch size. In this experiment we train the models using different batch sizes, adjusting the number of training steps accordingly to keep the total number of training points seen constant. With this experiment we aim to show the training behaviour of the different architectures if we go from single example batches (many, potentially noisier updates) to batches of size 128 - a batch size in which each batch contains in expectation several examples per case, but fewer updates are made to the network parameters. Besides replicating several insights made in the main text, this experiment shows: (1) smaller batches require a smaller learning rate, supporting our argument that hyper-parameters should not be optimized independent of each other. (2) The focus of *BERT* on the *first*-case when the learning rate is too high is amplified in smaller batches.

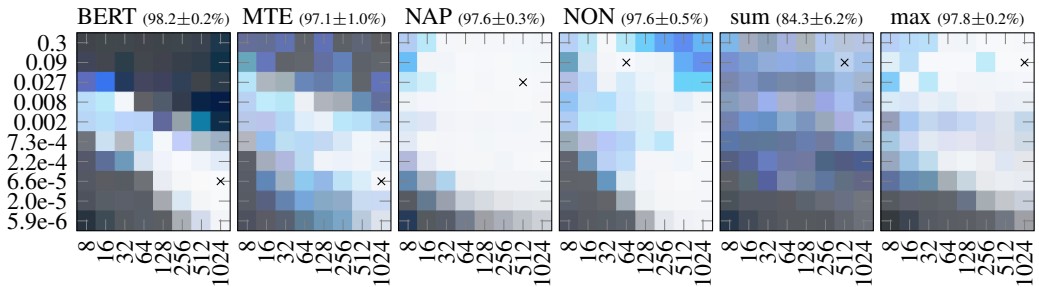

Figure 20: Rerun of the experiments of Figure 5 with $L = 6$ instead of $L = 2$. Crosses indicate the combination for best validation accuracy, which we report behind the model name.

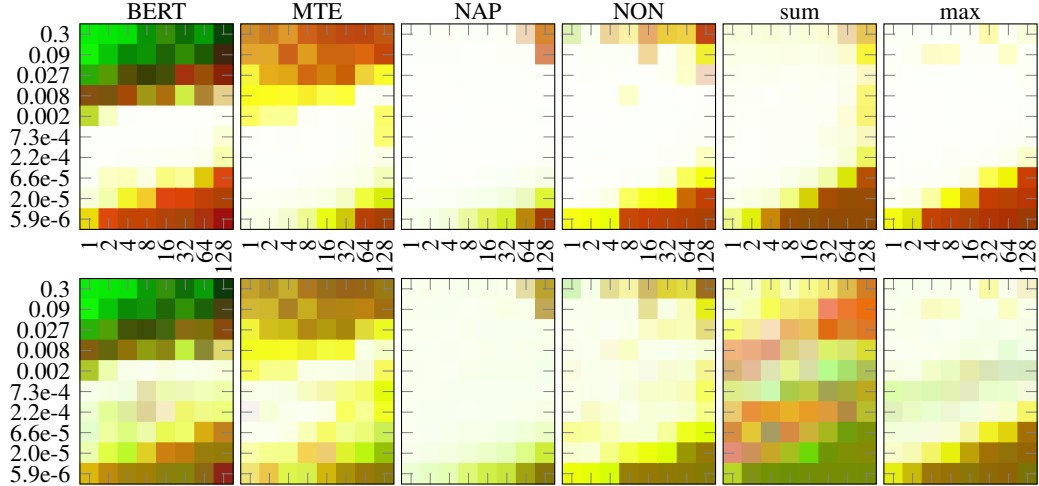

Figure 21: Learning rate (y-axis) vs. batch size (x-axis) on the argmin-first-argmax case distinction task (with output across all tokens). RGB pixel values correspond to *argmin-*, *first-* and *argmax-*case-accuracies, respectively. **Top row:** Training accuracy (sequence length $N = 128$). **Bottom row:** Validation accuracy on sequences of length $N = 64$.

### F.3 VARYING INITIALIZATION SCALE

In Figure 22 we provide the case accuracy results of an additional experiment, varying the initialization standard deviation of the truncated normal distribution with which we initialize the weight matrices. The results show additionally to the observations made previously that the softmax based models *BERT* and *MTE* struggle to learn if the initialization is too large. This is likely due to the fact that the softmax is already saturated if the initialization too large. Also, larger initializations seem to require a larger learning rate.

### F.4 LOCAL VS. GLOBAL FOCUS UNDER VARYING DATA BIAS

To investigate the local vs. global focus further, we conduct another experiment under varying data bias. Specifically we look at a task where each token has to output its own position (*identity*-case), if there is a 64 in the input sequence. If there is no 64, each token has to output the *argmin* position. The results when varying the percentage of *identity*-case data points are given in Figure 23. Red represents that only the *identity*-case is learned while turquoise represents that only the *argmin*-case is learned. We can see that *BERT* struggles to learn the *argmin*-case (which requires global information) as soon as it is not the majority case anymore.

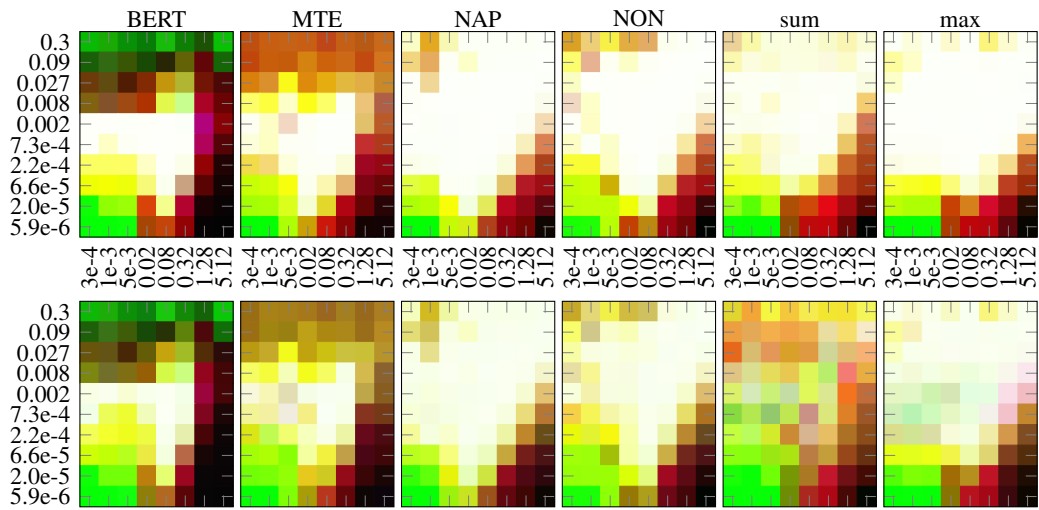

Figure 22: Learning rate (y-axis) vs. initialization scale (x-axis) on the case distinction task (output across tokens). RGB pixel values correspond to the case-accuracies. **Top row:** Training accuracy ($N = 128$). **Bottom row:** Validation ($N = 64$).

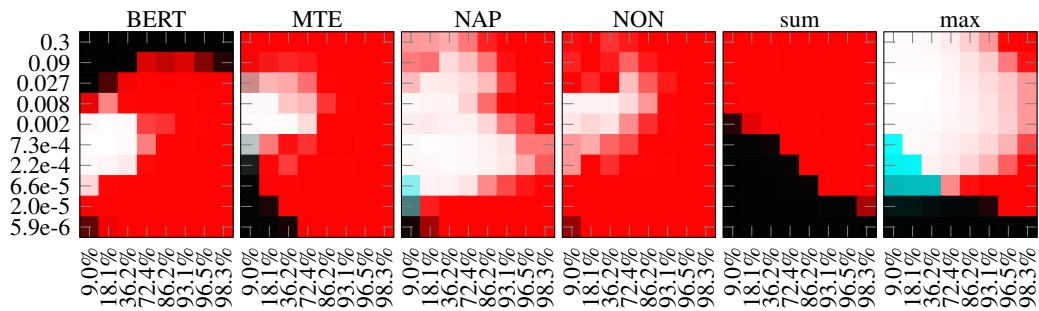

Figure 23: Learning rate (y-axis) vs. percentage of *identity*-case (x-axis). R = *identity* accuracy, G and B = *argmin* accuracy.

## F.5 FIRST TOKEN OUTPUT - VARYING MODEL DIMENSION

Section 5.2 discusses the case accuracies when training on the case distinction task with outputs taken from the first token. In Figure 24 we addtionally give best the min-, mean- and max-accuracies over the course of training. The top row corresponds to in-distribution/training accuracy ($N = 128$) while the bottom row corresponds to out-of-distribution generalization accrucay when validating on sequences of half the length ($N = 64$). Again we note a correlation between optimal learning rate and model dimension, especially in the *BERT* and *MTE* architecture. We also note that *BERT* and *MTE* have a large performance variation across random seeds in this setup.

## F.6 FIRST TOKEN OUTPUT - VARYING DEPTH

In this section we investigate whether our results are tied to the shallow architecture of $L = 2$ Transformer layers. We therefore vary the number of Transformer-layers $L$ and report the results on the case distinction task with outputs taken from the first token in Figure 25. The results lead us to the following observations: (1) The *BERT* architecture does seem to perform better when the number of Transformer-layers is increased to $L = 4$. However, the performance degrades if we further increase the depth. (2) The *NAP* architecture achieves a higher best mean accuracy and performs well on a wide range of depths. (3) The *max* architecture performs well on the biggest range of hyperparameters. This is due to the beneficial architectural prior as discussed earlier.

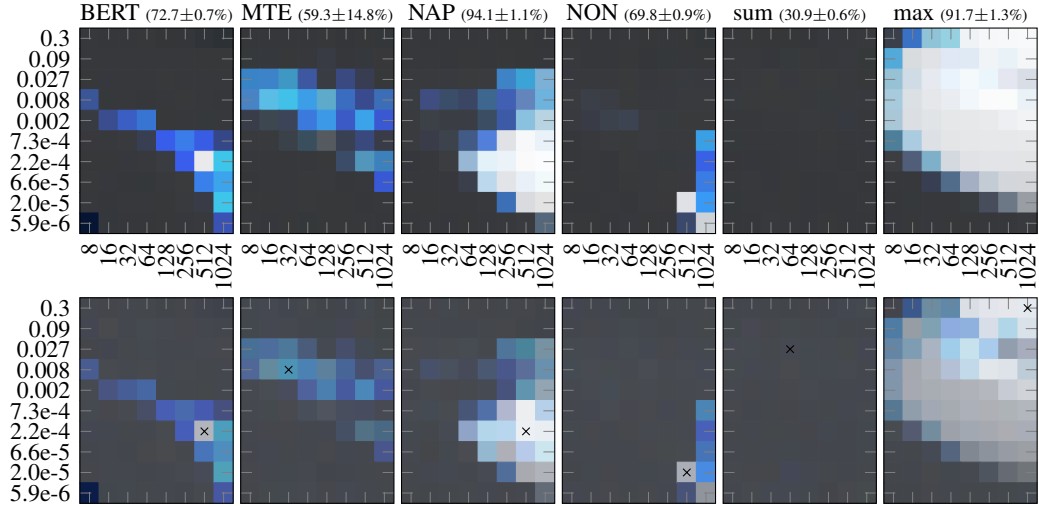

Figure 24: Learning rate (y-axis) vs. model dimension $d$ (x-axis) on the case distinction task with output from the first token. RGB pixel values correspond to min, mean and max accuracy. **Top row:** Training accuracy (sequence length $N = 128$). **Bottom row:** Validation accuracy when validating on sequences of half the length ($N = 64$). Crosses indicate the combination for best validation accuracy, which we report with standard deviation behind the model name.

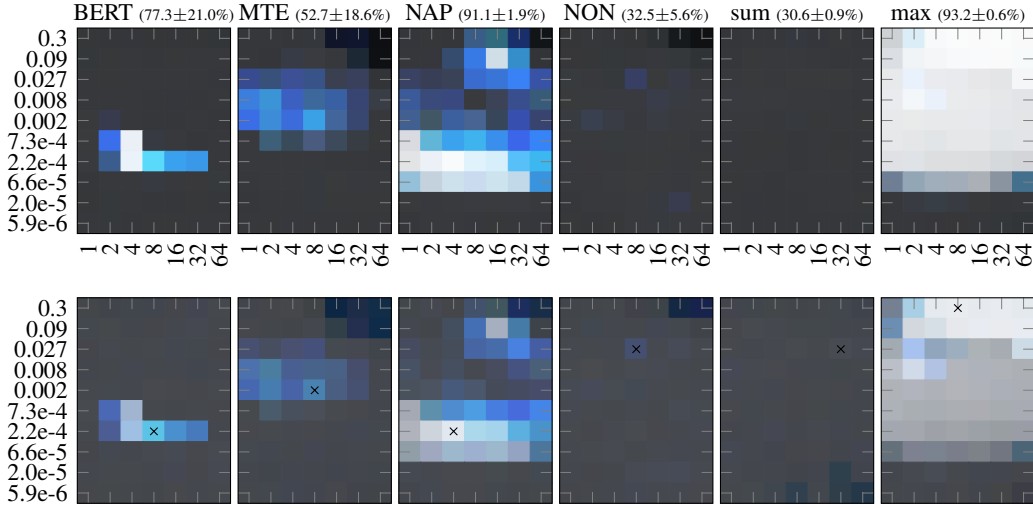

Figure 25: Learning rate (y-axis) vs. Transformer-layers $L$ (x-axis) on the case distinction task (output from the first token). RGB pixel values correspond to min, mean and max accuracy. **Top row:** Training accuracy (sequence length $N = 128$). **Bottom row:** Validation accuracy when validating on sequences of half the length ($N = 64$). Crosses indicate the combination for best validation accuracy, which we report with standard deviation behind the model name.

# G  MODE FINDING TASK - ADDITIONAL RESULTS

## G.1  VARYING VOCABULARY SIZE

Figure 26 shows the results of an additional experiment, varying the vocabulary size $S$ while keeping the sequence length $N = 128$ constant during training. For this experiment, we also vary the total number of training steps and set it to $400 \cdot S$, to keep the number of examples seen per vocabulary token approximately constant. We also include zero-shot generalization results when testing on

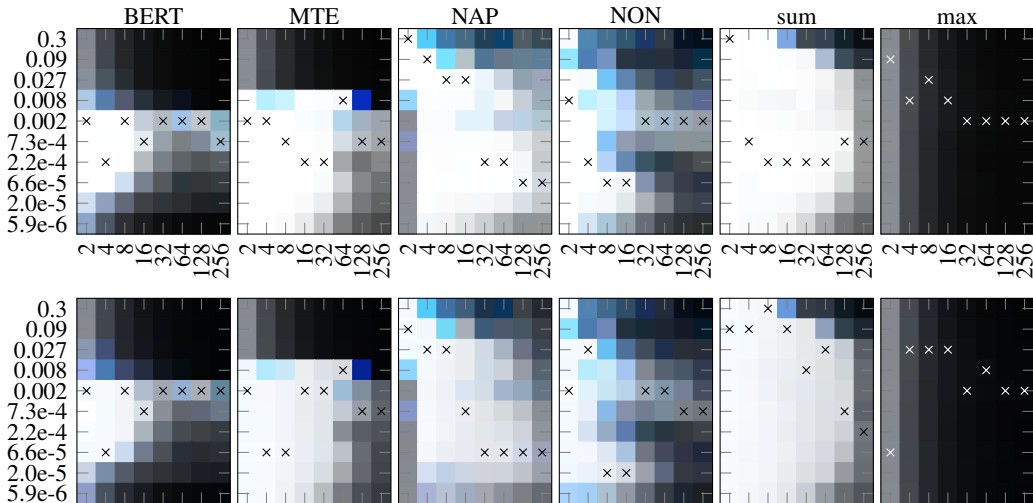

Figure 26: Learning rate (y-axis) vs. vocabulary size $S$ (x-axis) on the mode finding task. RGB pixel values correspond to min, mean and max accuracy. **Top row:** Training accuracy (sequence length $N = 128$). **Bottom row:** Validation accuracy when validating on sequences of twice the length ($N = 256$). Crosses indicate the learning rate for best mean accuracy, reported in Table 5.

Table 5: Best mean accuracy per vocabulary size, taken from the combinations indicated in Figure 26. First six rows correspond to training accuracies, bottom six rows correspond to validation accuracies. Bold numbers indicate a min-accuracy higher than the best max accuracy of all other models.

|      | $S = 2$ | $S = 4$ | $S = 8$ | $S = 16$ | $S = 32$ | $S = 64$ | $S = 128$ | $S = 256$ |
|------|---------|---------|---------|----------|----------|----------|-----------|-----------|
| BERT | 100%    | 99.9%   | 99.9 %  | 92.1%    | 72.5%    | 76.2%    | 77.4%     | 74.4%     |
| MTE  | 100%    | 100%    | 99.9%   | 99.8%    | 99.3%    | 97.3%    | 73.3%     | 64.9%     |
| NAP  | 100%    | 99.9%   | 99.8%   | 99.6%    | 99.7%    | **99.6%** | 97.4%    | **84.6%** |
| NON  | 100%    | 99.9%   | 99.2%   | 97.3%    | 74.7%    | 71.5%    | 65.2%     | 61.5%     |
| sum  | 100%    | 100%    | 99.9%   | 99.8%    | 99.7%    | 99.2%    | 97.5%     | 60.6%     |
| max  | 55.7%   | 30.1%   | 17.3%   | 10.4%    | 6.6%     | 4.6%     | 3.6%      | 3.1%      |
| BERT | 100%    | 98.2%   | 95.8 %  | 88.0%    | 65.7%    | 68.6%    | 68.0%     | 53.0%     |
| MTE  | 99.2%   | 98.4%   | 96.1%   | 93.6%    | 90.5%    | 85.4%    | 61.6%     | 38.9%     |
| NAP  | 99.6%   | 98.4%   | 95.8%   | 93.1%    | 90.6%    | 90.4%    | 84.3%     | **64.3%** |
| NON  | 100%    | 97.7%   | 93.1%   | 85.7%    | 66.4%    | 58.3%    | 50.2%     | 46.4%     |
| sum  | 99.0%   | 97.9%   | 96.7%   | 94.4%    | 91.6%    | 89.1%    | 85.9%     | 45.6%     |
| max  | 53.8%   | 29.3%   | 16.1%   | 9.5%     | 6.0%     | 4.2%     | 3.0%      | 2.1%      |

sequences of twice the length ($N = 256$). Compared to the case distinction task we can do such a generalization evaluation here as we do not learn any positional embeddings in this setup. We make the following observations: (1) *max* completely fails to learn in any of the vocabulary sizes. Note that the shading to the left merely corresponds to the majority class base rate. (2) *NAP* struggles when the vocabulary consists of only 2 tokens. This is expected, as the mean subtraction in the normalization effectively removes the task relevant information (the mode) in this case. Note however, that for a high enough learning rate, the model learns to use the bias parameter $b$ introduced in the normalization - effectively reverting to sum pooling. (3) While all models learn the task well on small vocabularies, *NAP* outperforms all other approaches significantly when $S$ gets larger then the training sequence length, cf. Table 5.

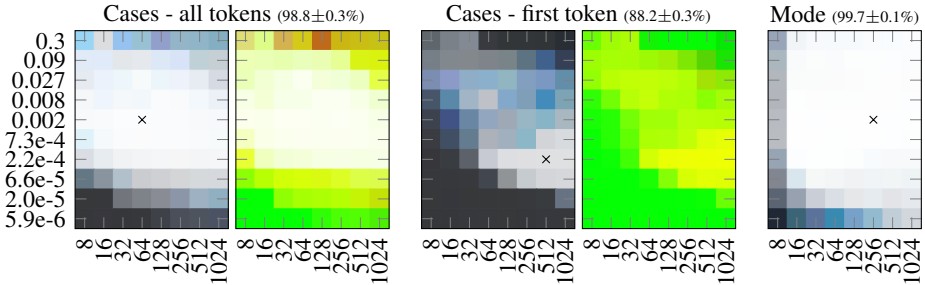

Figure 27: Learning rate (y-axis) vs. model dimension $d$ (x-axis) in the different task setups with learned aggregation weights. The plots from left to right are grouped according to the tasks: The case task with outputs across all tokens (cf. Section 5.1), the case task with outputs from the first token (cf. Section 5.2) and the mode task (cf. Section 5.3). For the case tasks, the left sub-plot reports min, mean and max accuracies while the right sub-plot reports mean case accuracies as RGB pixel values. The mode plot reports min, mean and max accuracies. The plots show the accuracies for $N = 128$.

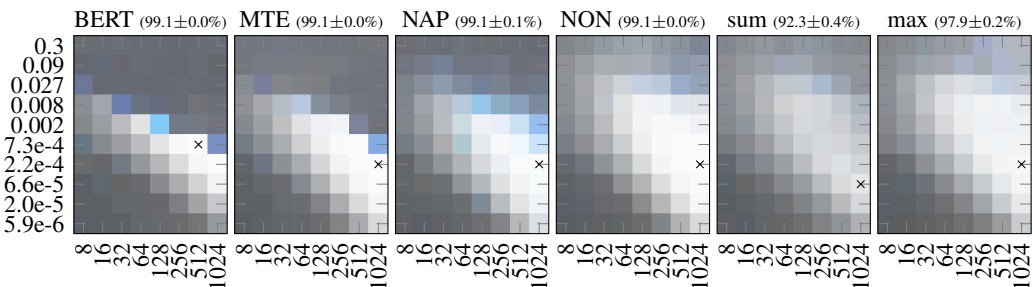

Figure 28: Full results of the GNN experiments on the protein-protein-interaction task. The plot shows a variation of learning rate (y-axis) vs. model dimension $d$ (x-axis). RGB pixel values correspond to min, mean and max validation node classification F1-score. Crosses indicate the best combination (validation), from which we report the test performance behind the model name.

## H  LEARNED AGGREGATION WEIGHTS

Here we present the results of an additional model for fixed aggregation size. In particular, we ask how the results look if we learn the aggregation weights as parameters of the network. For this model, we take the *sum* model and replace the sum-broadcast operation with a fully connected linear layer across the sequence dimension. This is akin to the Random Synthesizer presented by Tay et al. (2021), but without the softmax. The results presented in Figure 27 confirm our conjecture that the softmax limits the information flow, as this new model is more robust to hyperparameters than *BERT* and *MTE*.

## I  PPI EXPERIMENT

For the PPI node classification task we train a GNN based on our architectures with aggregation over neighbors instead of the whole sequence. Here we use $L = 3$ transformer layers, as the number of layers specifies the $k$-hop neighborhood from which information is aggregated in a GNN ($k = L$). The remaining settings are left at the default ($M = 4$ attention heads, no regularization). We train with one graph per update for 32 epochs.

All runs are repeated with 5 different random seeds. We track the validation performance over the course of training. Figure 10 shows the highest validation performance in each hyperparameter combination, with the results of all models given in Figure 28. For the test performance reported

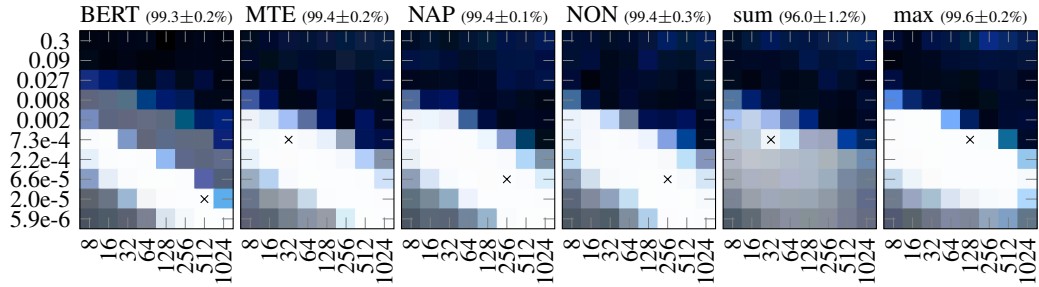

Figure 29: Full results of the RL experiments. The plot shows a variation of learning rate (y-axis) vs. model dimension $d$ (x-axis). RGB pixel values correspond to min, mean and max success rate when evaluating on 10,000 newly generated games. Crosses indicate the combination for best success rate, which we report with standard deviation behind the model name. Note here that training is stopped early if a success rate of 99% is reached.

we take the hyperparameter combination with the best average validation performance and use the checkpoints with highest validation performance over the course of training.

## J   BABY AI EXPERIMENT

To test our architecutres in an RL environment, we base ourself on the open source implementation of the working memory graph agent presented by Loynd et al. (2020). Specifically, we take a factored representation of the environment and let the agent learn, what it needs to remember. Please refer to Loynd et al. (2020) for the core ideas here. In our experiments, we slightly modify their code, replacing the multi layer perceptron for the policy and value outputs with a simple linear layer each. Also, we replace the ReLU non-linearities used with GeLUs. These modifications are done to reflect a similar setup as investigated in the rest of the paper. Similarly we fix $M = 4$, $L = 2$ and set the "WMG Memo size" equal to the model dimension $d$. We chose level 3 of the Baby-AI environment (Chevalier-Boisvert et al., 2019) based on the results presented by Loynd et al. (2020). That is, we chose a task which trains decently fast and which should be solvable for a variety of hyperparameter combinations given the allocated training time. For the hyperparameters which do not alter the architecture, we used those which Loynd et al. (2020) reported to work best for their architecture. For completeness, we list them in Table 6. The complete result of all architectures investigated are given in Figure 29. Also here we ran every hyperparameter combination with 5 different random seeds.

Table 6: Hyperparameters used in the reinforcement learning experiment. These are based on the hyperparameters which Loynd et al. (2020) report in their Appendix in Table 8.

| Hyperparameter | Value |
|---|---|
| A3C $t_{max}$ | 1 |
| Adam eps | 1e-12 |
| Discount factor $\gamma$ | 0.95 |
| Entropy term strength $\beta$ | 0.1 |
| Gradient clipping threshold | 128.0 |
| Reward scale factor | 8.0 |
| WMG Memos | 2 |

## K    Broader Impact Statement

Our work contrast different architectures on an abstract level. Due to the abstract nature of the study, there is no direct risk associated with system failure. However, we would like to mention potential implications of the findings. Our main proposal is an architecture which is more robust to hyperparameters. This has a potential positive benefits on the environment as less computation needs to be invested to reach a decent performance. Further, our architecture shows an increased robustness to skewed data distributions. This can have negative and positive aspects. On one hand, outliers inconsistent with ethical norms, such as hate speech and extreme views, might get picked up by the model more easily. On the other hand, the model might be able to represent minority groups more accurately, even though they are underrepresented in the data. In general, we see it as vital to get a better understanding of how different architectures learn to represent different biases in the data and hope that our work can provide a stepping stone in this direction.

## L    Figure Replication

Since our way to visualize the results uses the full color spectrum, it might be difficult for people with color blindness to verify and learn from our findings. We therefore replicate the main figures here with the R, G and B channels split into three separate plots. We also encourage readers to get in contact with us if they face any other difficulty reading our plots. We replicate Figures 5-10 in Figures 30-36 and the Figures of the Appendix thereafter. We also invite an interested reader to check out our code release, which allows for further visualizations of all results.

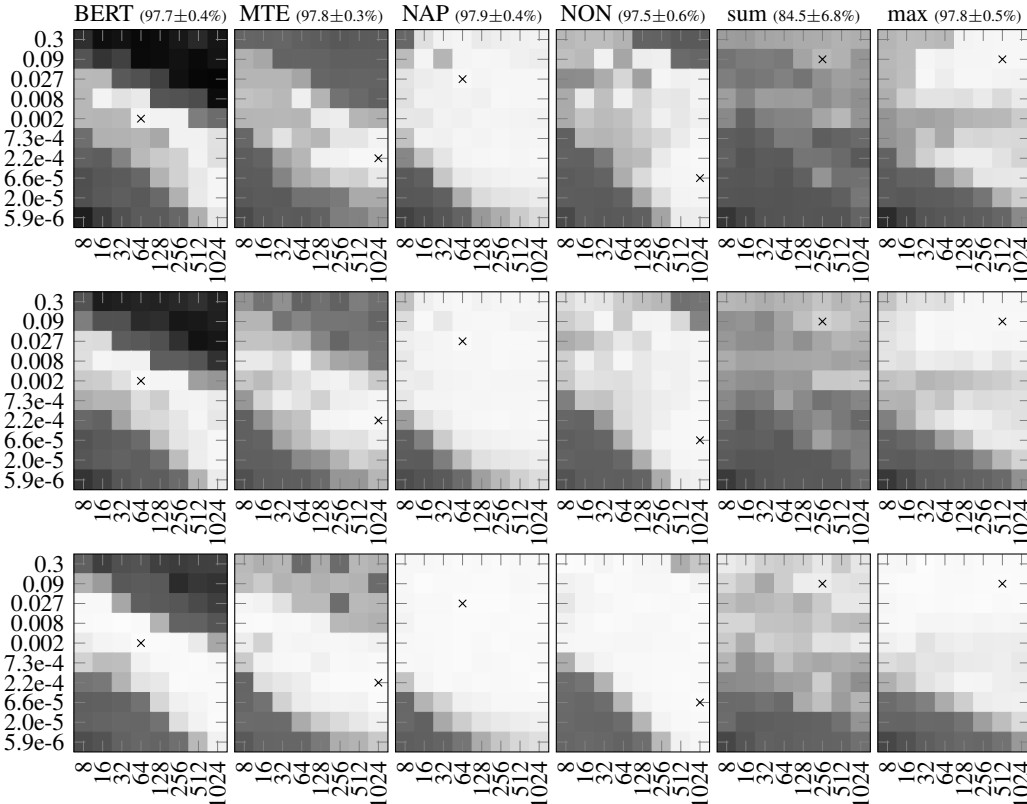

Figure 30: Replica of Figure 5: Learning rate (y-axis) vs. model dimension $d$ (x-axis) on the argmin-first-argmax case distinction task (with output across all tokens). The rows from top to bottom correspond to min-, mean- and max-accuracy, respectively. The plots show the validation accuracy when validating on sequences of length $N = 64$. Crosses indicate the combination for best validation accuracy, which we report with standard deviation behind the model name.

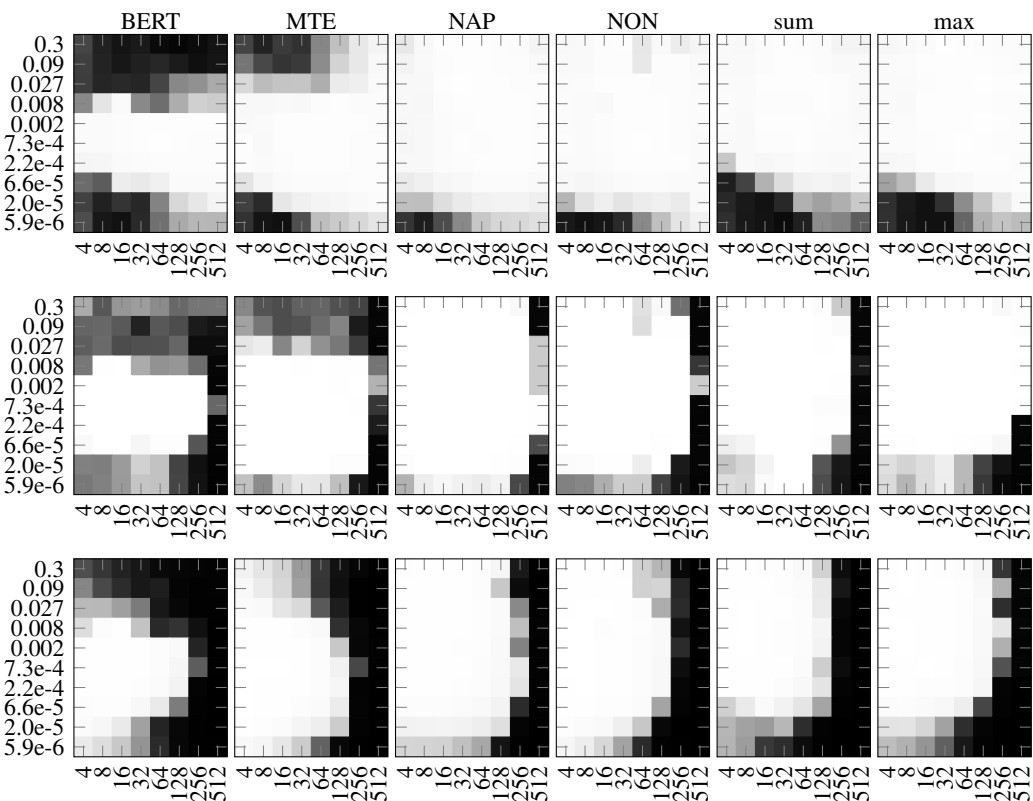

Figure 31: Replica of Figure 6 (top row): Biased data results on the case distinction task (with output across all tokens). The rows from top to bottom correspond to *argmin*, *first* and *argmax*-mean-case-accuracies, respectively. Shown is the learning rate (y-axis) vs. sequence length $N$ (x-axis).

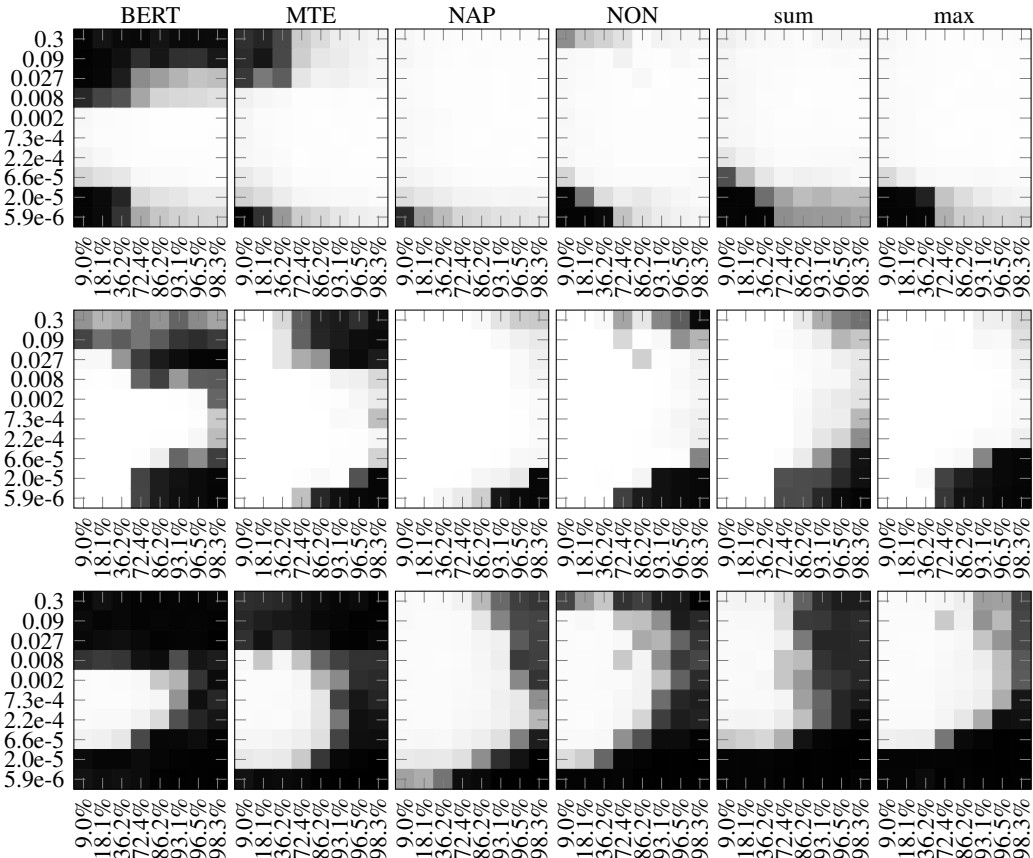

Figure 32: Replica of Figure 6 (bottom row): Biased data results on the case distinction task (with output across all tokens). The rows from top to bottom correspond to *argmin*, *first* and *argmax*-mean-case-accuracies, respectively. Shown is the learning rate (y-axis) vs. percentage of *argmin*-case in the data (x-axis) with fixed $N = 128$.

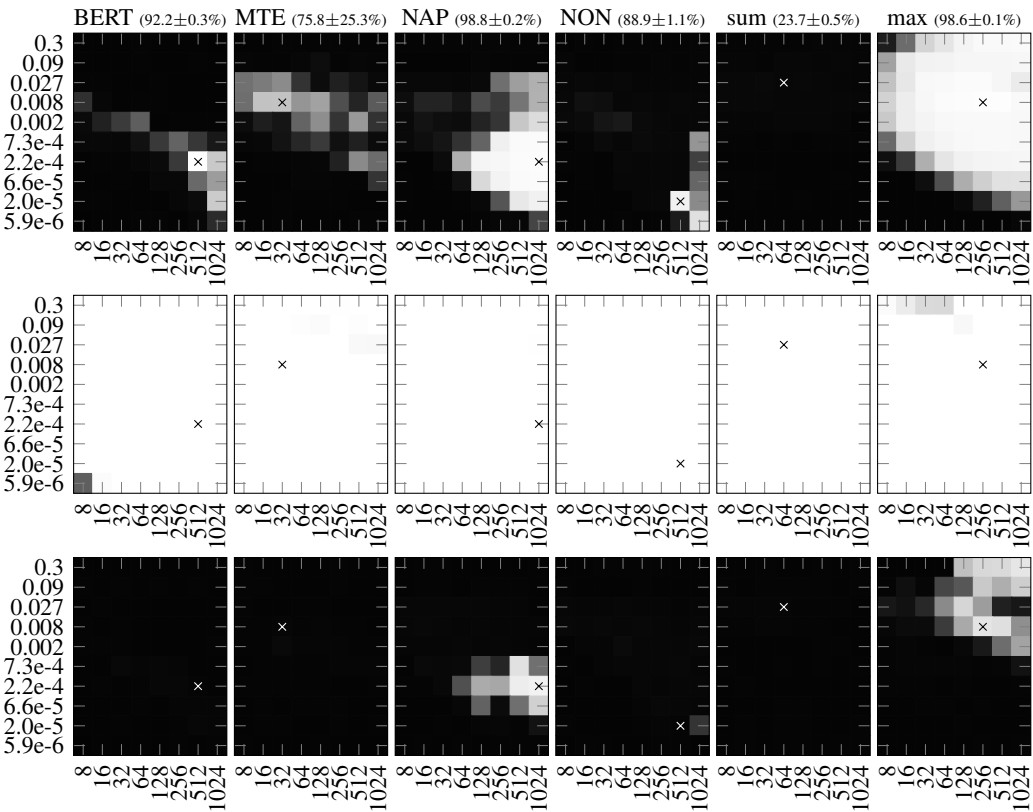

Figure 33: Replica of Figure 7: Learning rate (y-axis) vs. model dimension $d$ (x-axis) on the case distinction task with output from the first token. The rows from top to bottom correspond to *argmin*, *first* and *argmax*-mean-case-accuracies. Crosses indicate the best accuracy, reported behind the model name.

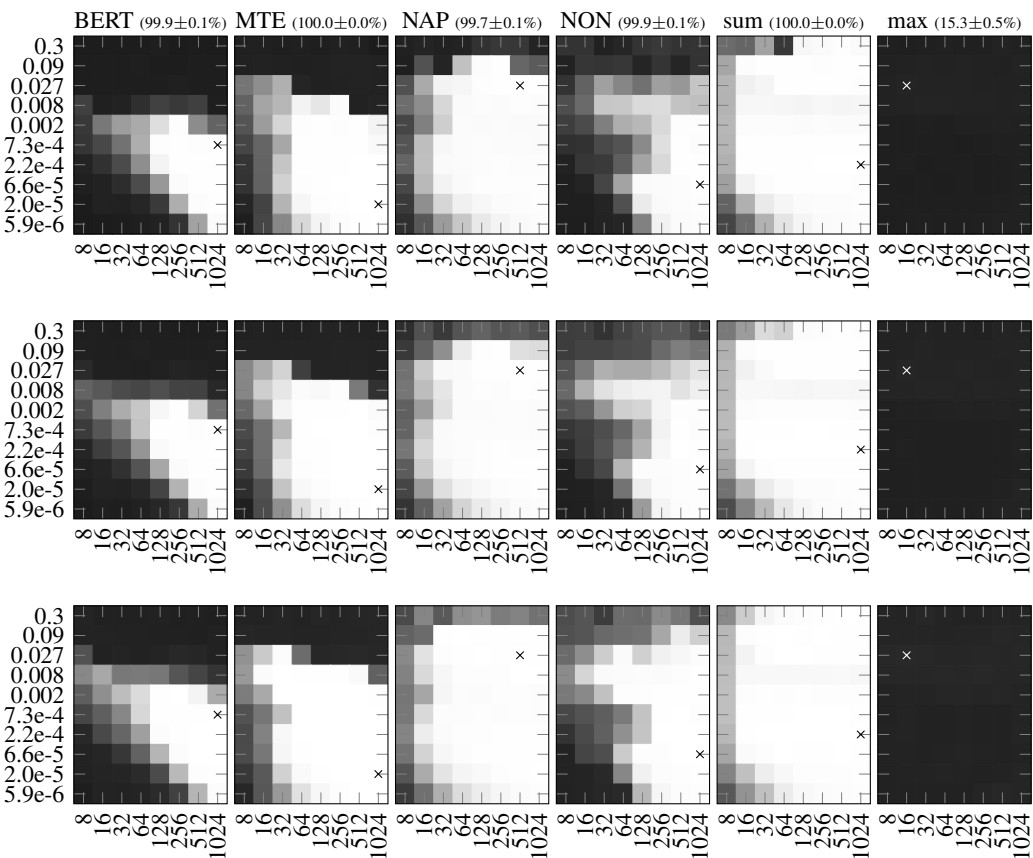

Figure 34: Replica of Figure 8: Learning rate (y-axis) vs. model dimension $d$ (x-axis) on the mode finding task. The rows from top to bottom correspond to min, mean and max accuracy. Crosses indicate the combination for best accuracy, reported behind the model name.

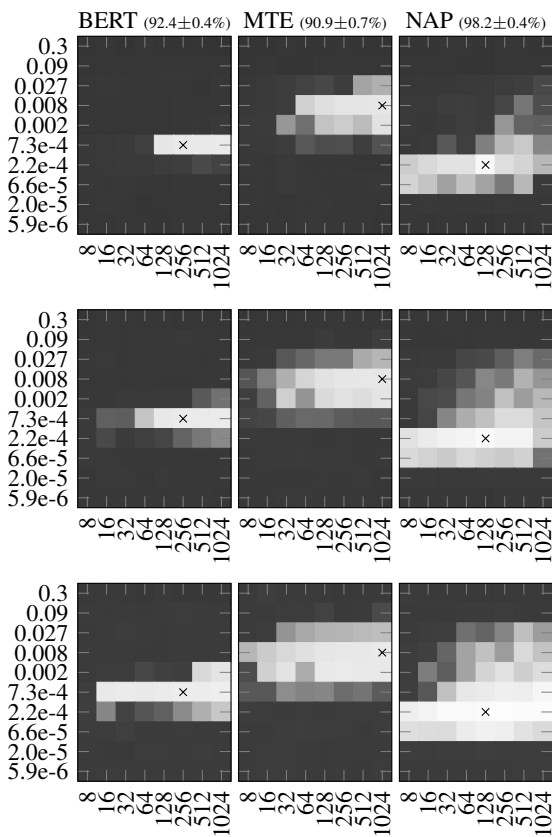

Figure 35: Replica of Figure 9: Learning rate (y-axis) vs. attention heads $M$ (x-axis) on the case distinction task (first token output). The rows from top to bottom correspond to min, mean and max accuracy. Crosses indicate the combination for best accuracy, reported behind the model name.

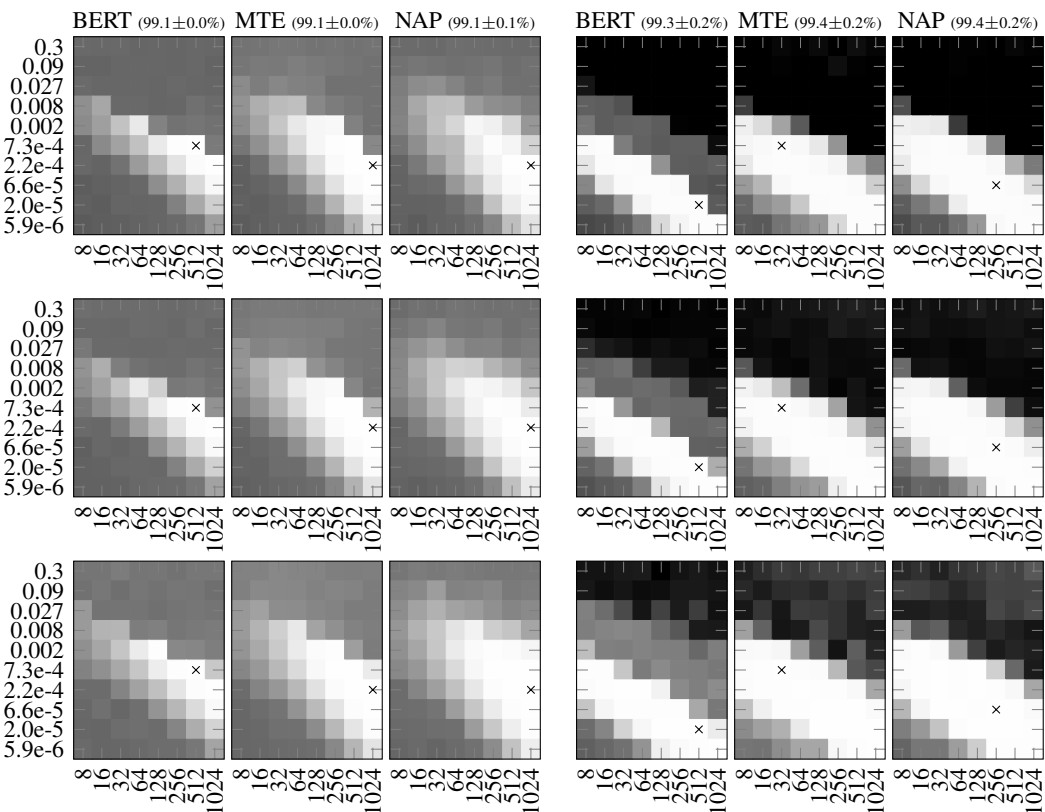

Figure 36: Replica of Figure 10: Learning rate (y-axis) vs. model dimension $d$ (x-axis). The rows from top to bottom correspond to min, mean and max validation performance. Crosses indicate the best combination (validation), from which we report the test performance. **Left:** Protein-protein-interaction task. Shown is the node classification F1-score. **Right:** Altered working memory graph agent in the Baby-AI level 3 reinforcement learning task. Shown is the success rate.

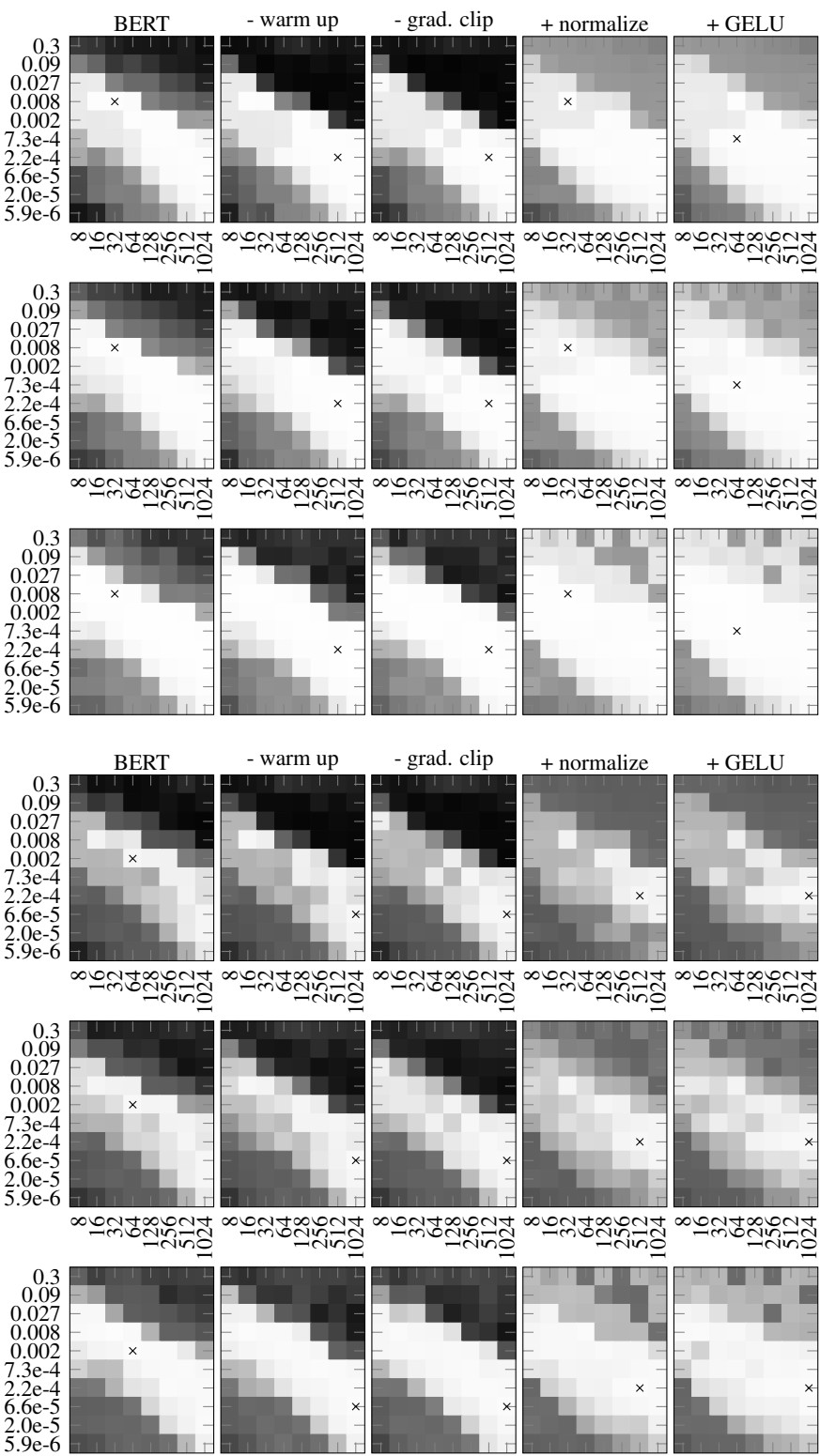

Figure 37: Replica of Figure 13 (top two rows): Learning rate (y-axis) vs. model dimension $d$ (x-axis) on the case distinction task (with output across all tokens) - ablation study. The rows from top to bottom correspond to min, mean and max accuracy for $N = 128$ (top 3 rows) and $N = 64$ (bottom 3 rows). Crosses indicate the combination for best accuracy, reported in Table 1.

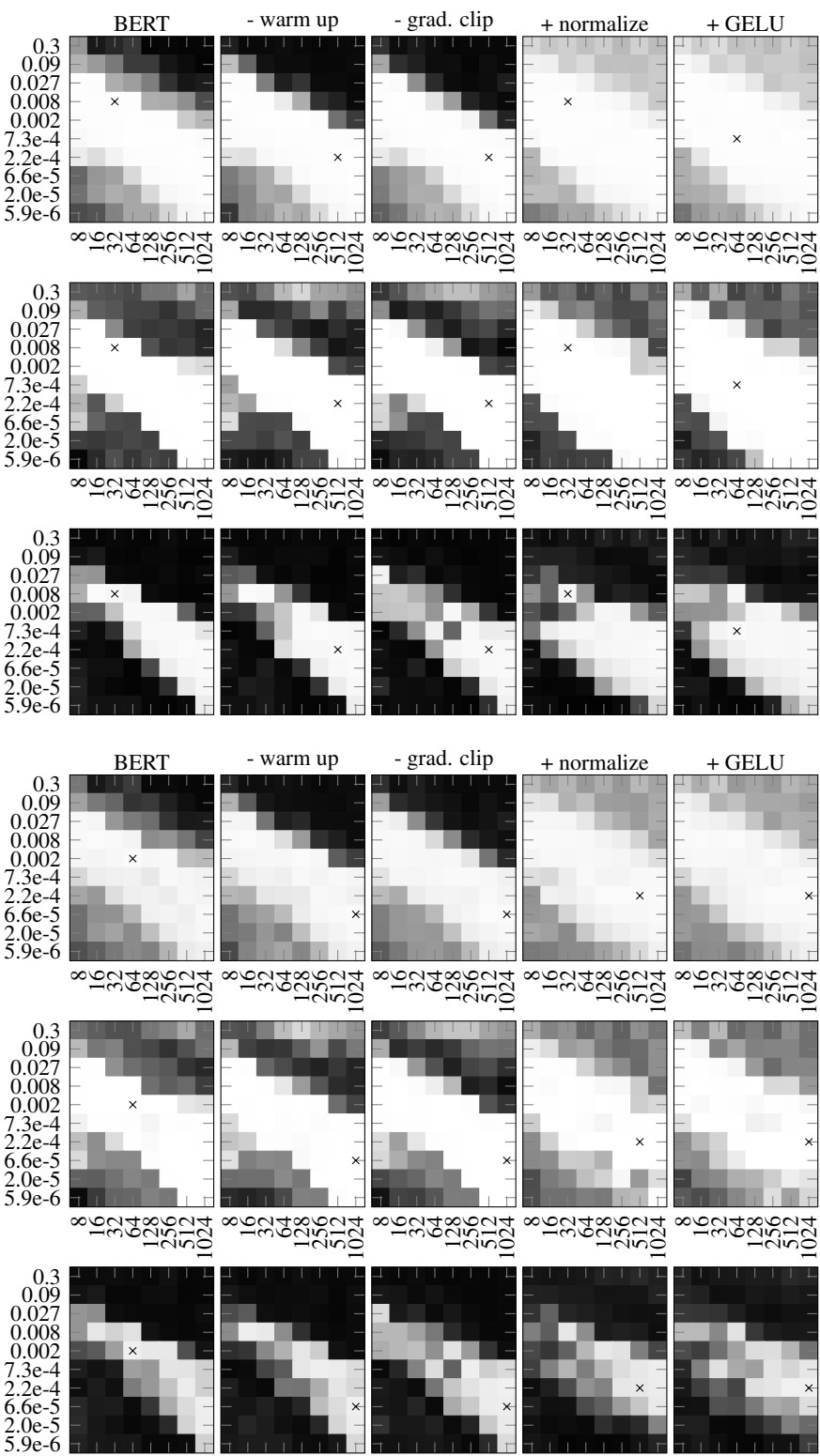

Figure 38: Replica of Figure 13 (bottom two rows): Learning rate (y-axis) vs. model dimension $d$ (x-axis) on the case distinction task (with output across all tokens) - ablation study. The rows from top to bottom correspond to *argmin*, *first* and *argmax* accuracy for $N = 128$ (top 3 rows) and $N = 64$ (bottom 3 rows). Crosses indicate the combination for best accuracy, reported in Table 1.

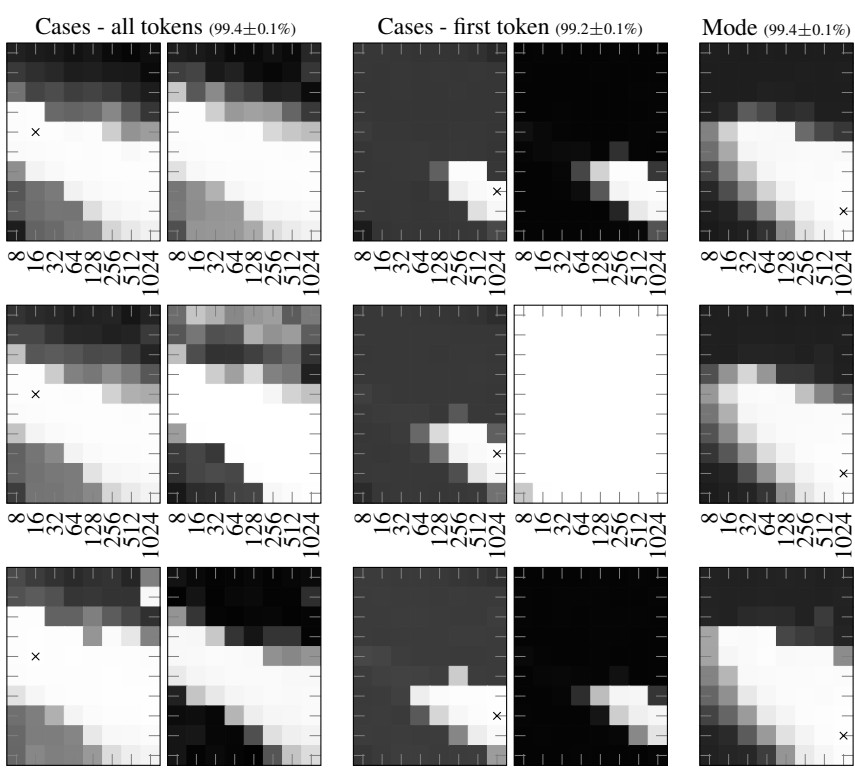

Figure 39: Replica of Figure 14: Learning rate (y-axis) vs. model dimension $d$ (x-axis) in the different task setups when just replacing the softmax in *BERT* with normalization. The plots from left to right are grouped according to the tasks: The case task with outputs across all tokens (cf. Section 5.1), the case task with outputs from the first token (cf. Section 5.2) and the mode task (cf. Section 5.3). For the case tasks, the left sub-plots report min, mean and max accuracies (from top to bottom) while the right sub-plots report mean case accuracies (*argmin*, *first* and *argmax* from top to bottom). The mode plots report min, mean and max accuracies. The plots show the accuracies for $N = 128$.

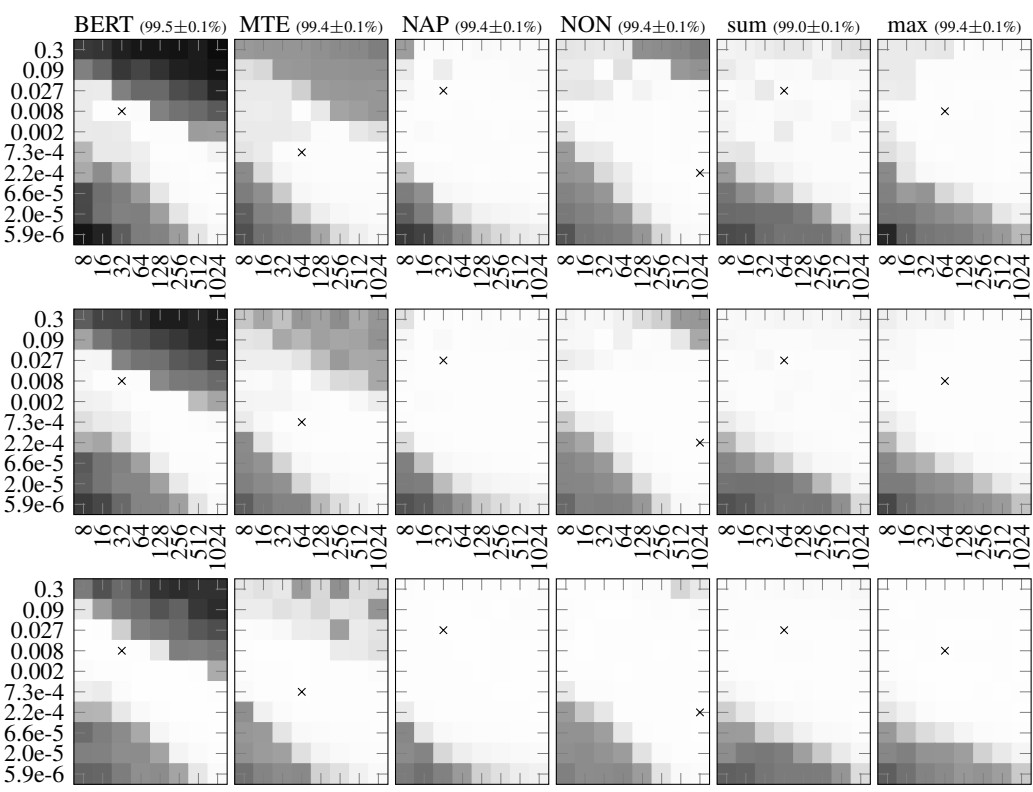

Figure 40: Replica of Figure 18: Learning rate (y-axis) vs. model dimension $d$ (x-axis) on the argmin-first-argmax case distinction task (with output across all tokens). The rows from top to bottom show min, mean and max accuracy for $N = 128$. Crosses indicate the combination for best accuracy, reported behind the model name.

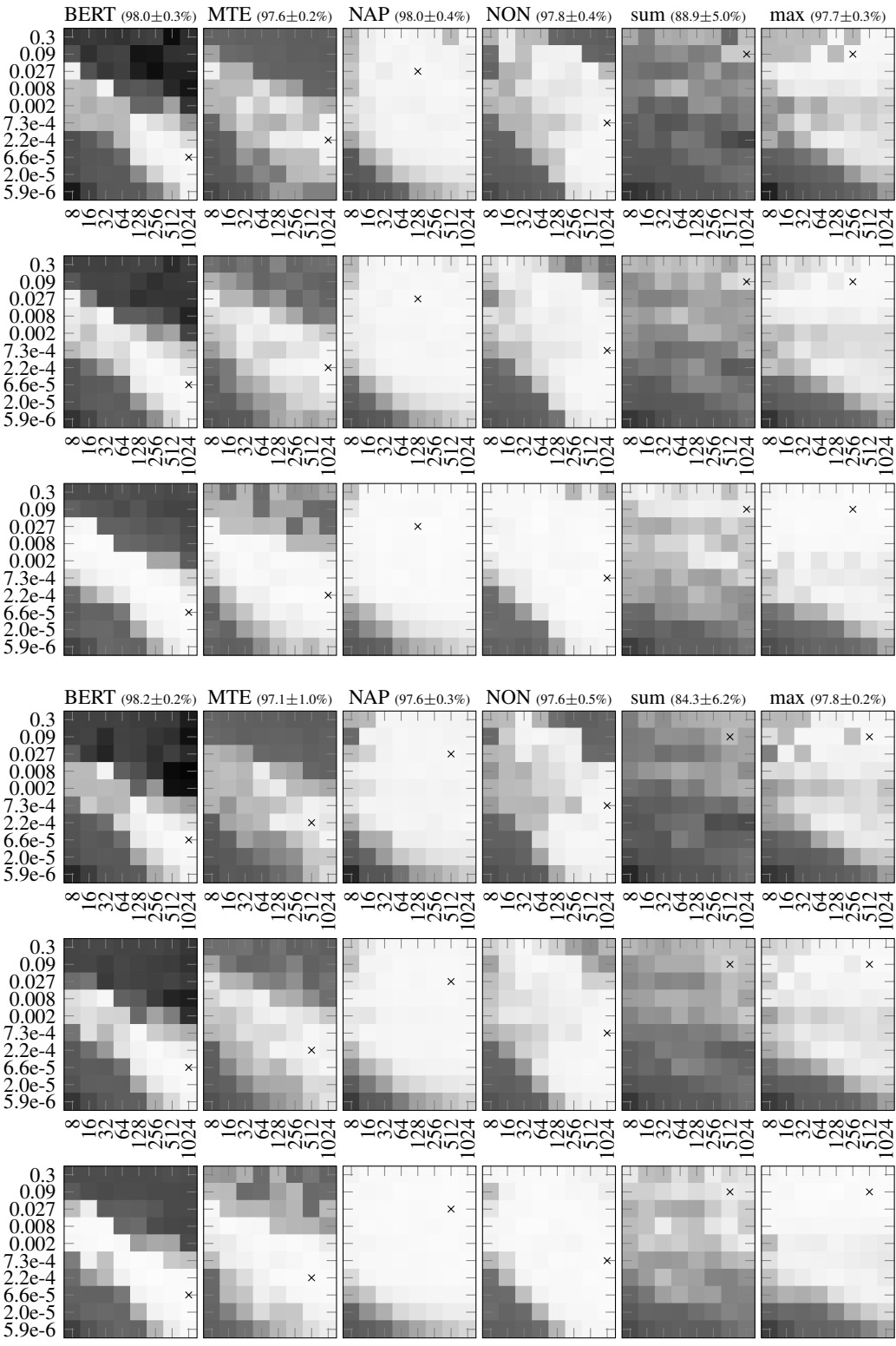

Figure 41: Replica of Figures 19 and 20: Learning rate (y-axis) vs. model dimension $d$ (x-axis) on the argmin-first-argmax case distinction task (with output across all tokens) with $L = 4$ (top 3 rows) and $L = 6$ (bottom 3 rows). The rows from top to bottom show min, mean and max accuracy for $N = 64$. Crosses indicate the combination for best accuracy, reported behind the model name.

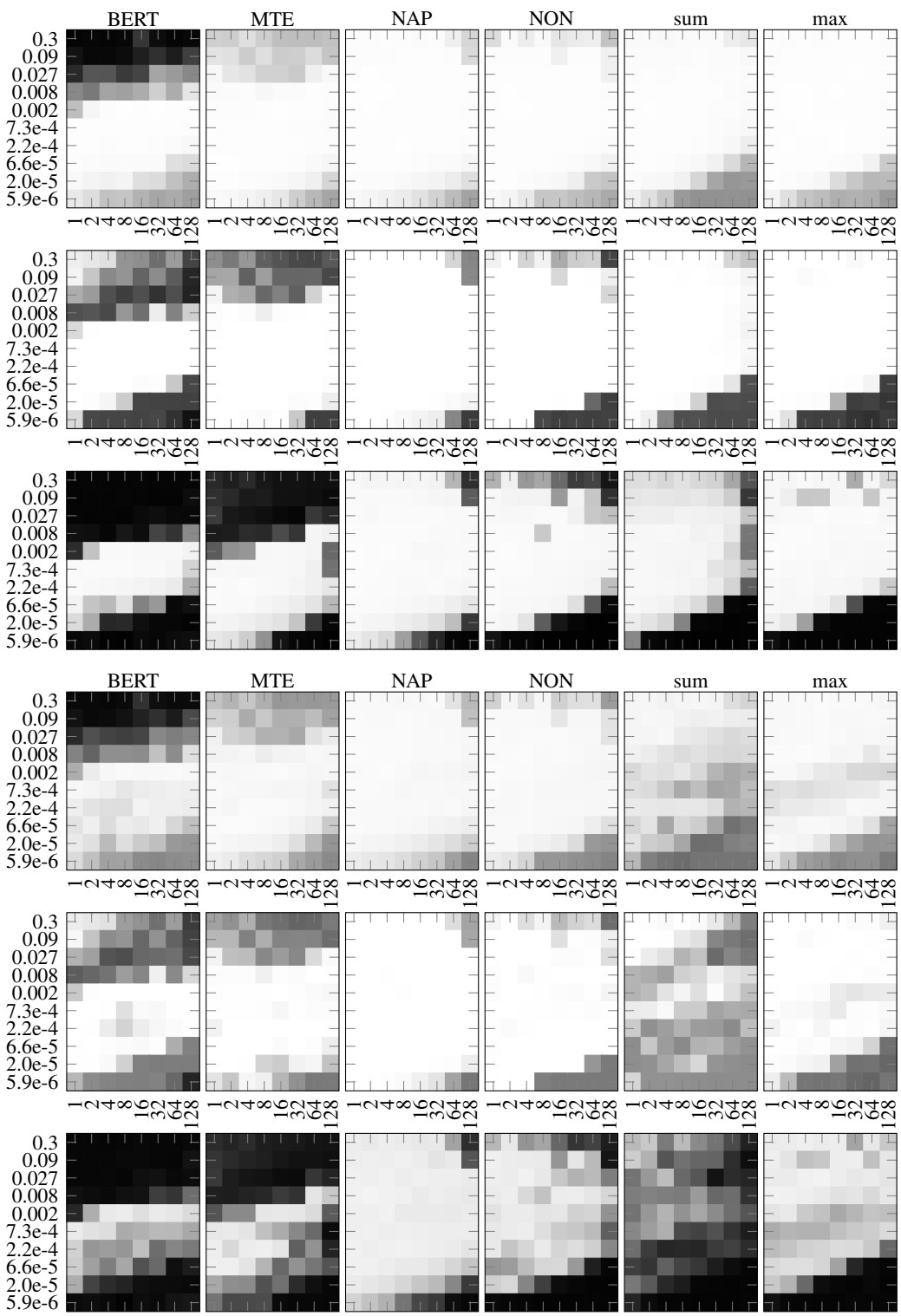

Figure 42: Replica of Figure 21: Learning rate (y-axis) vs. batch size (x-axis) on the case distinction task (with output across all tokens). The rows from top to bottom correspond to *argmin*, *first* and *argmax*-mean-case-accuracies for $N = 128$ (top 3 rows) and $N = 64$ (bottom 3 rows). Crosses indicate the combination for best accuracy, reported behind the model name.

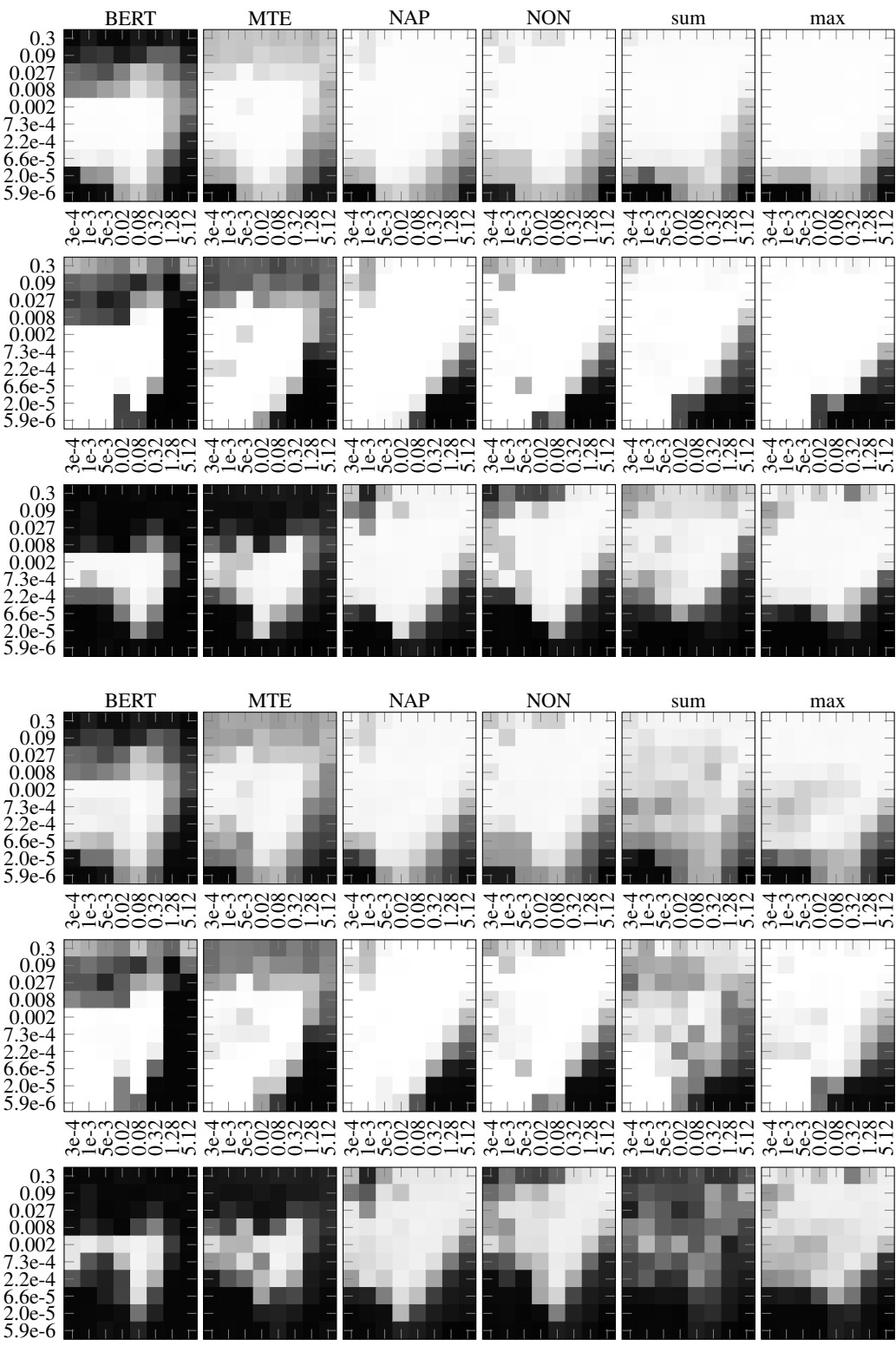

Figure 43: Replica of Figure 22: Learning rate (y-axis) vs. initialization scale (x-axis) on the case distinction task (output across tokens). The rows from top to bottom correspond to *argmin*, *first* and *argmax*-mean-case-accuracies for $N = 128$ (top 3 rows) and $N = 64$ (bottom 3 rows).

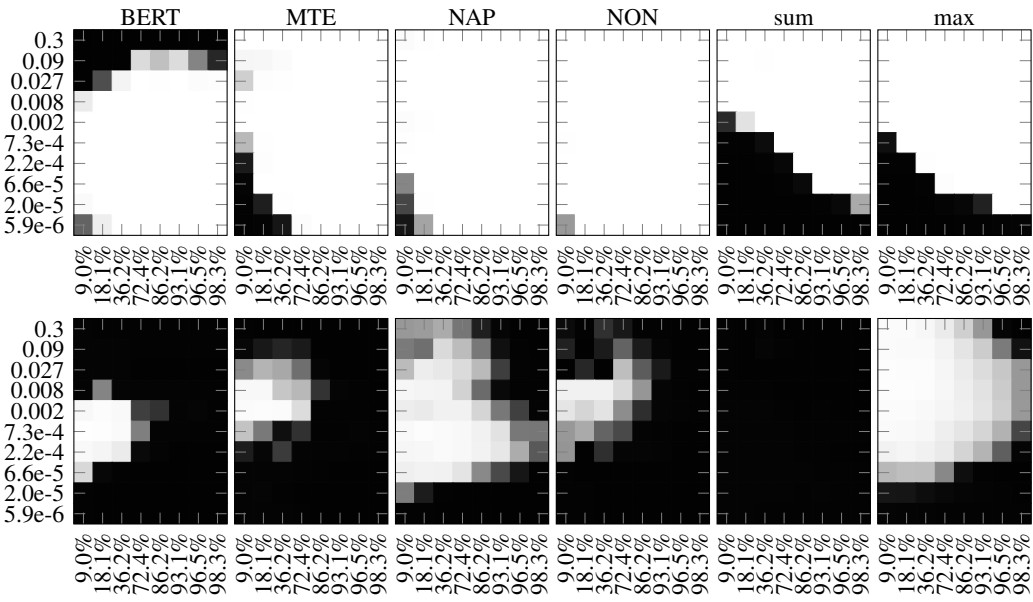

Figure 44: Replica of Figure 23: Learning rate (y-axis) vs. percentage of *identity*-case (x-axis). **Top row:** *identity* accuracy. **Bottom row:** *argmin* accuracy.

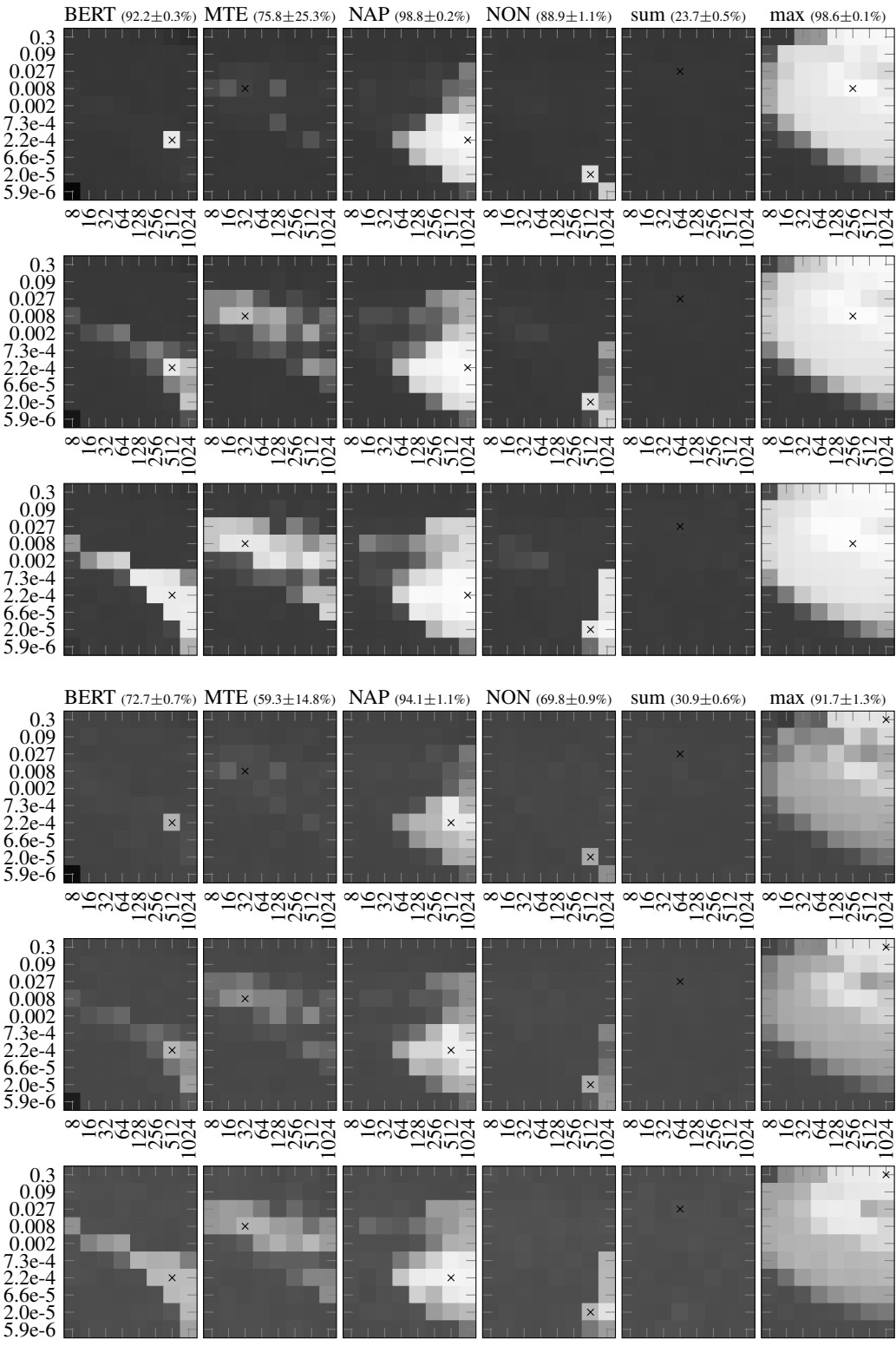

Figure 45: Replica of Figure 24: Learning rate (y-axis) vs. model dimension $d$ (x-axis) on the case distinction task (first token output). The rows from top to bottom show min, mean and max accuracy for $N = 128$ and then $N = 64$. Crosses indicate the combination for best accuracy, reported behind the model name.

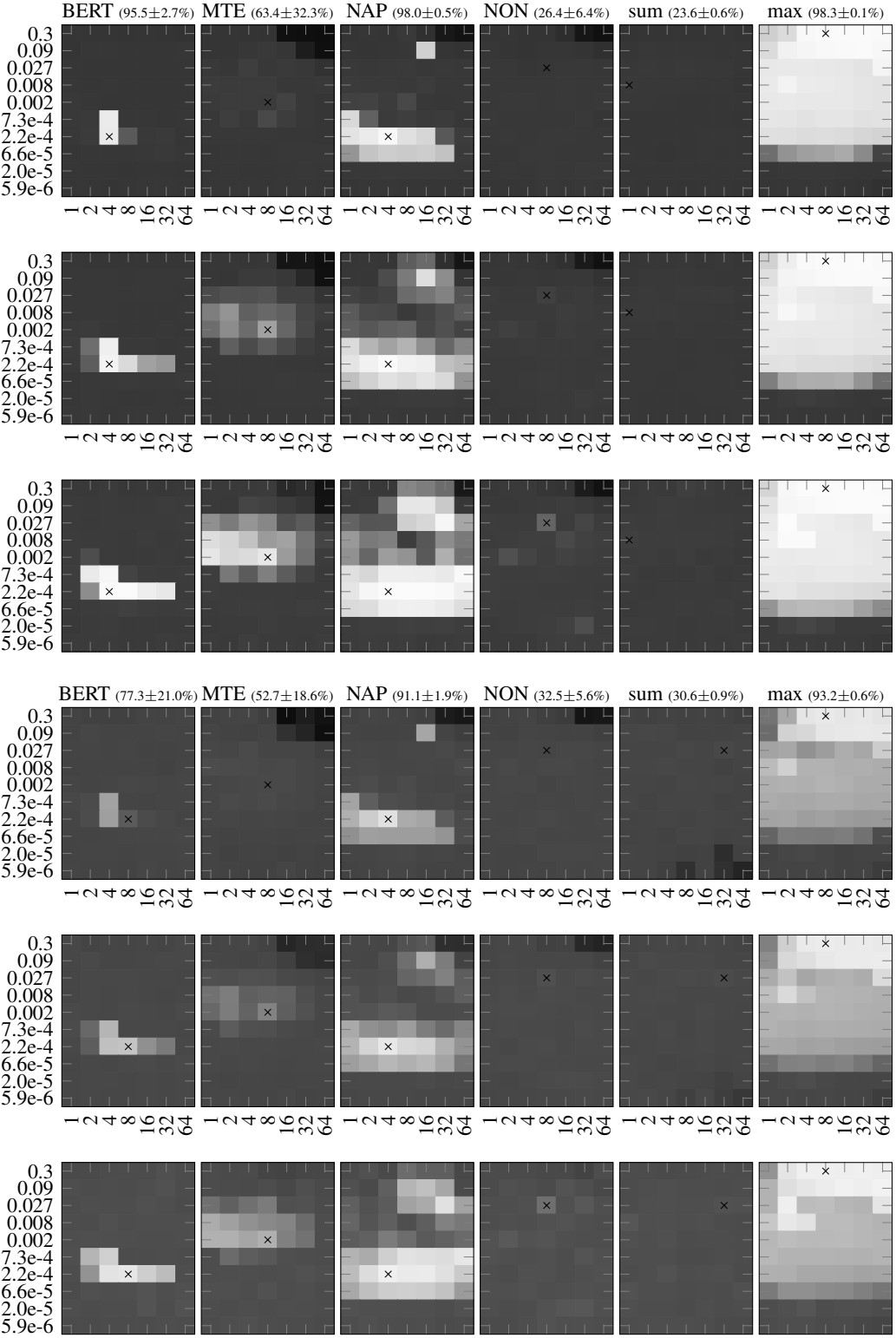

Figure 46: Replica of Figure 25: Learning rate (y-axis) vs. Transformer-layers $L$ (x-axis) on the case distinction task (output from the first token). The rows from top to bottom correspond to min, mean and max accuracy for $N = 128$ (top 3 rows) and $N = 64$ (bottom 3 rows). Crosses indicate the combination for best mean accuracy, reported behind the model name.

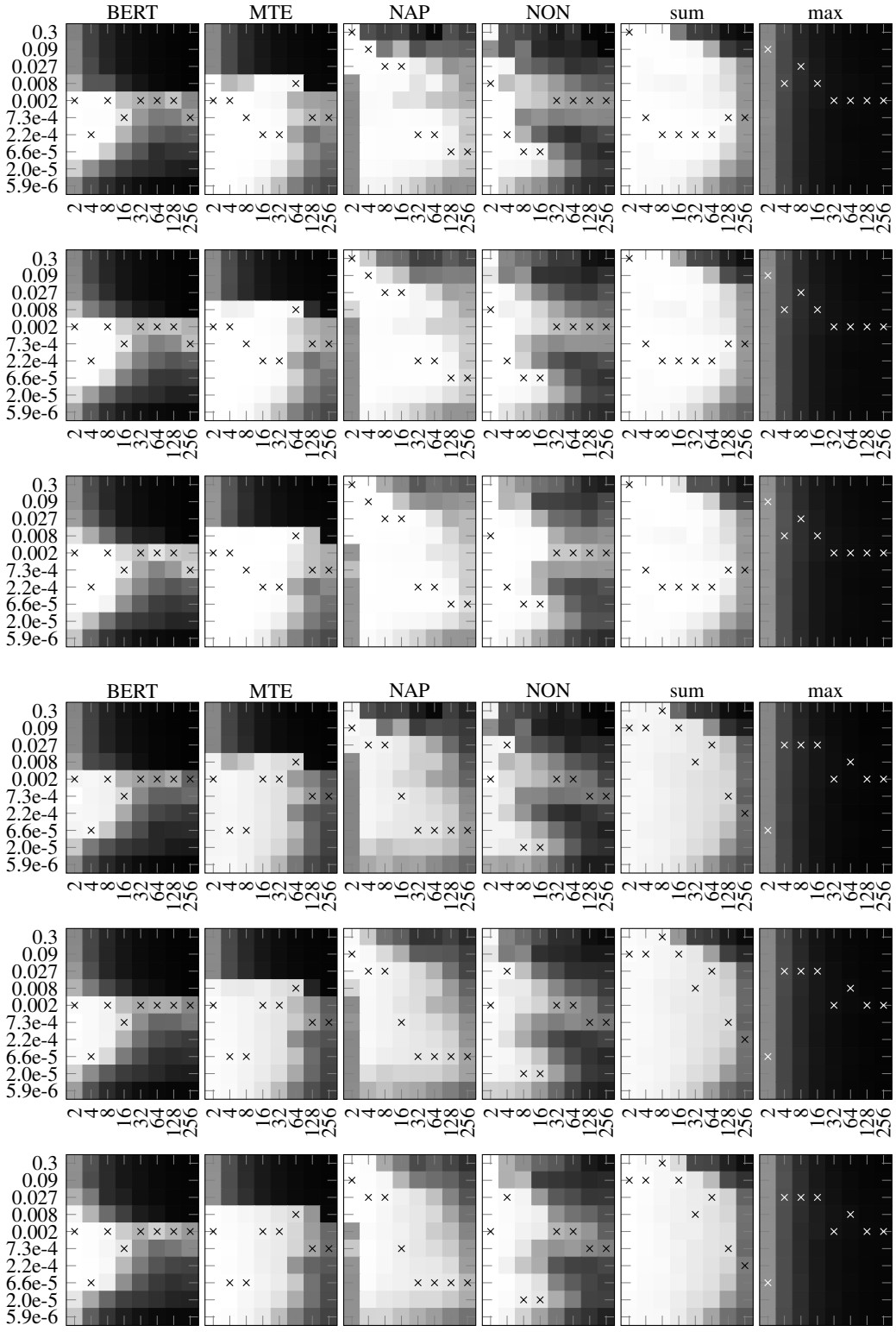

Figure 47: Replica of Figure 26: Learning rate (y-axis) vs. vocabulary size $S$ (x-axis) on the mode finding task. The rows correspond to min, mean and max accuracy for $N = 128$ (top 3 rows) and $N = 256$ (bottom 3 rows). Crosses indicate the learning rate for best mean accuracy, reported in Table 5.

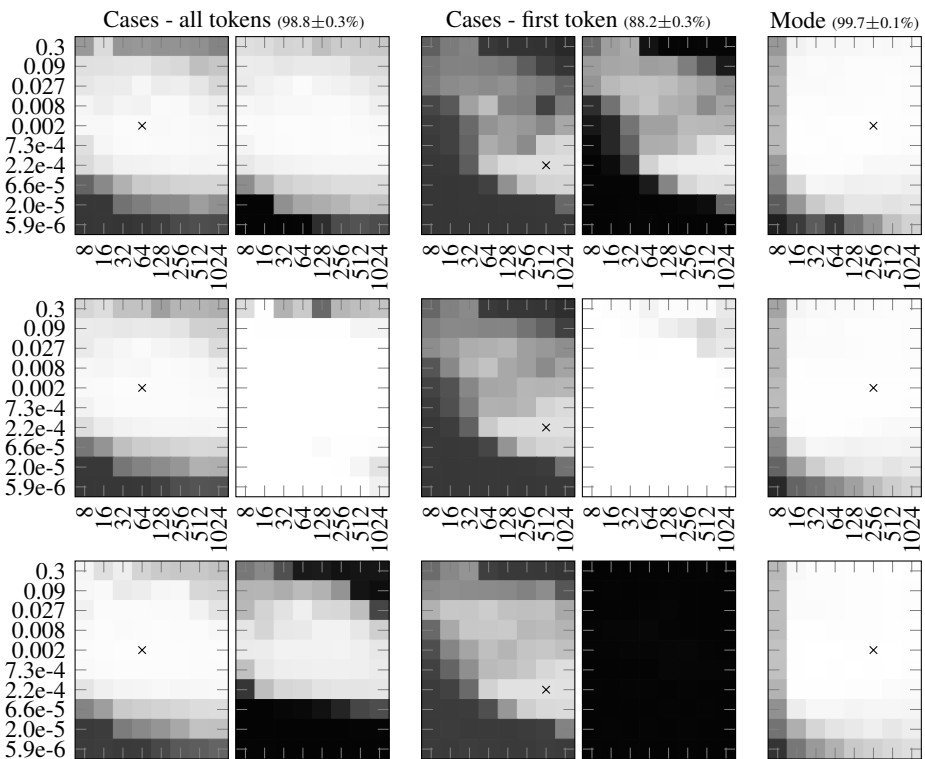

Figure 48: Replica of Figure 27: Learning rate (y-axis) vs. model dimension $d$ (x-axis) in the different task setups with learned aggregation weights. The plots from left to right are grouped according to the tasks: The case task with outputs across all tokens (cf. Section 5.1), the case task with outputs from the first token (cf. Section 5.2) and the mode task (cf. Section 5.3). For the case tasks, the left sub-plots report min, mean and max accuracies (from top to bottom) while the right sub-plots report mean case accuracies (*argmin*, *first* and *argmax* from top to bottom). The mode plots report min, mean and max accuracies. The plots show the accuracies for $N = 128$.

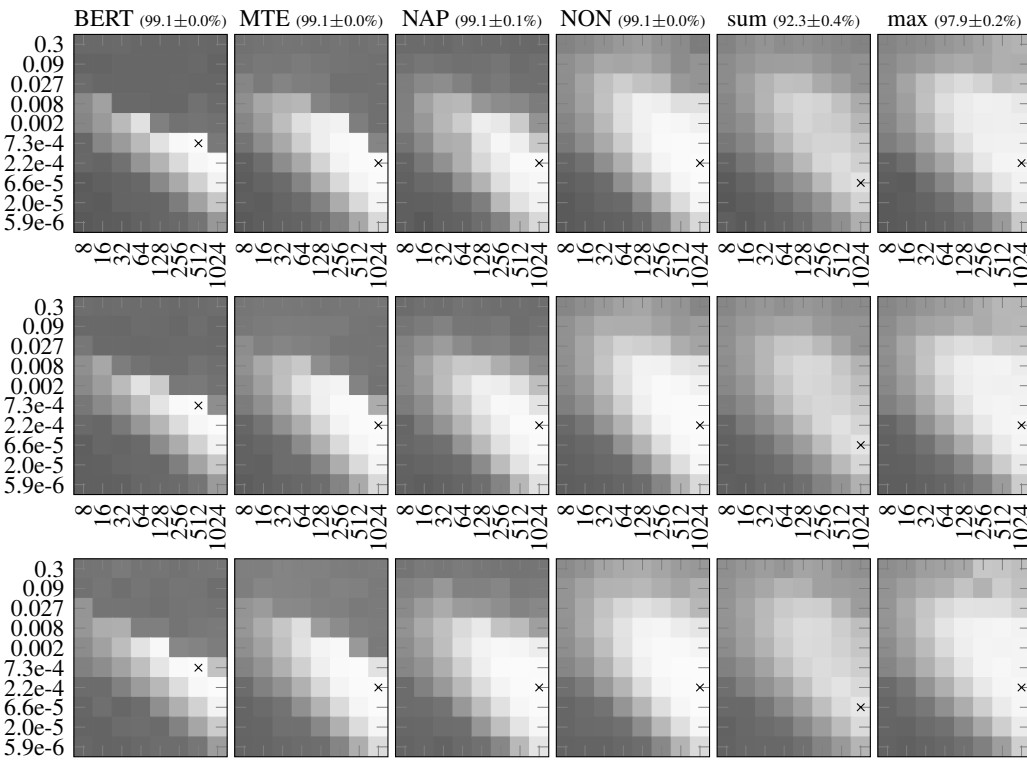

Figure 49: Replica of Figure 28: Full results of the GNN experiments on the protein-protein-interaction task. The plot shows a variation of learning rate (y-axis) vs. model dimension $d$ (x-axis). The rows from top to bottom correspond to min, mean and max validation node classification F1-score. Crosses indicate the best combination (validation), from which we report the test performance behind the model name.

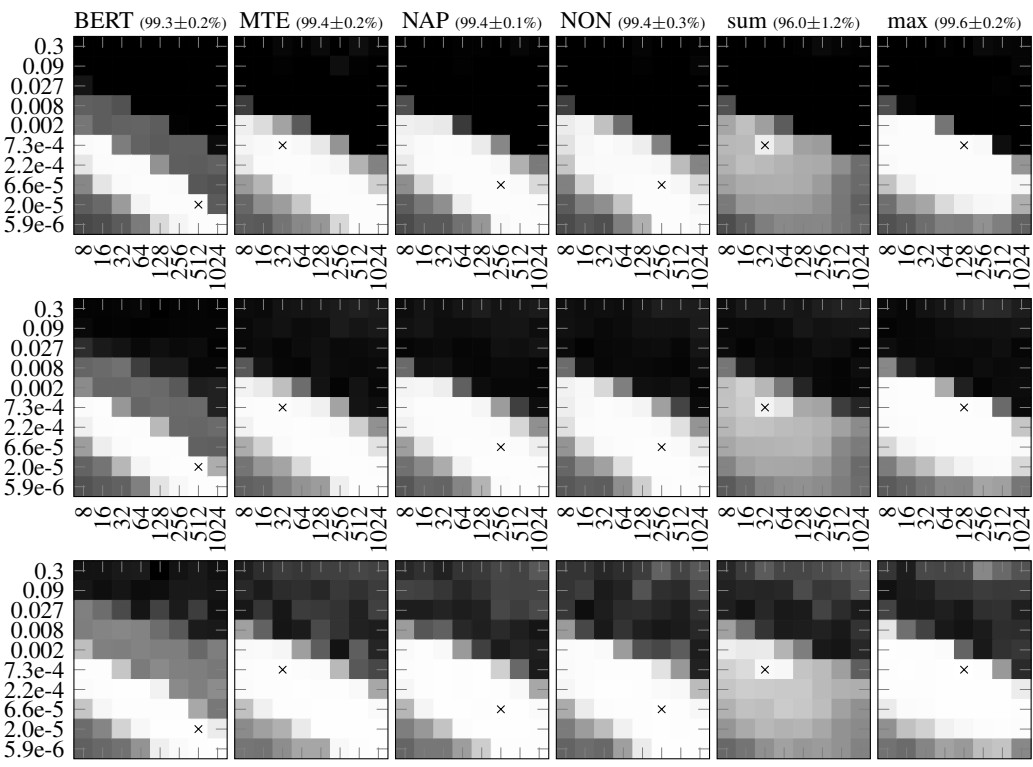

Figure 50: Replica of Figure 29: Full results of the RL experiments. The plot shows a variation of learning rate (y-axis) vs. model dimension $d$ (x-axis). The rows from top to bottom correspond to min, mean and max success rate when evaluating on 10,000 newly generated games. Crosses indicate the combination for best success rate, which we report with standard deviation behind the model name. Note here that training is stopped early if a success rate of 99% is reached.

