# OpenReview forum: "Normalized Attention Without Probability Cage"
_ICLR.cc/2022/Conference — ICLR 2022 Submitted_

### Official Review · Reviewer_BH5E · 2021-11-02

**Correctness:** 3
**Technical Novelty And Significance:** 4
**Empirical Novelty And Significance:** 4
**Recommendation:** 8
**Confidence:** 4

**Main Review:**

Pros:
1. The paper is clearly written and easy to follow.
2. The implications of using softmax attention are thoroughly analyzed in theoretical and experimental settings.
3. The set of synthetic experiments conducted highlights the differences between attention mechanisms reasonably well and supports the authors’ claims.

Cons:
1. The authors do not apply their proposed mechanism to the established BERT benchmarks. Instead, they construct a set of synthetic tasks that aim to imitate real-world NLP tasks. Lack of real-world NLP benchmarks makes it harder to believe that the implications of the softmax use actually matter. As the authors mentioned in Appendix C, some softmax-attention constraints may actually be beneficial, and this can’t be confirmed or disproven without real world benchmarks. The synthetic tasks chosen are intuitively similar to the specific NLP problems, but in the end, it’s not clear how well the synthetic results will generalize.
2. The experimental results demonstrate the advantage of NAP over sum and max pooling, but the case of NAP vs NON is less obvious. For many tasks, NON performs adequately well in terms of final quality and hyperparams sensitivity (in figure 26 it seemingly outperforms NAP for sensitivity). The paper would benefit from providing clear evidence (in addition to Figure 7) that despite all the additional layer norms in MTE, the dynamic normalization of attention logits is still necessary.

Suggestions/questions:
1. It’d be great to add the learned aggregation weights mechanism into the comparison. The authors explicitly try to avoid depending on the sequence length but having such a comparison may help prove the hypothesis that convex limitations are to blame and moving away from the convex hull is beneficial. For example, Figure 7 may benefit from such comparison.
2. Due to the use of MTE architecture, it’s not clear how well NAP will perform as a drop-in replacement of the softmax attention. The paper would benefit from NAP experiments that are not MTE-based to confirm that NAP isn’t brittle and can be used in practice.
3. (Figure 7) The explanation here can be more convincing. Why is MTE worse than traditional BERT on this task? Is it because local information bias is actually beneficial? Why is NON underperforming here and how can it be confirmed (NON doesn’t underperform like this in other tasks)? Figure 23 from the Appendix is more convincing, so it may make sense to move it here
4. (Figure 9) It’d be great to move the NON, sum, and max results from the appendix here and add the explanation.
5. (Style) Using RGB pixels for visualization purposes helps with the presentation but is not colorblind friendly. . It may be better to split the plots per-channel.


**Summary Of The Paper:**

The authors explore the implications of using softmax to implement attention mechanisms in NLP models, particularly transformers. They highlight the theoretical limitations of using softmax-attention and propose a normalization-based attention mechanism (NAP) that overcomes these constraints.. The authors conduct a set of synthetic experiments to compare different attention mechanisms’ performance and demonstrate that the proposed mechanism performs well compared to the traditional softmax attention while being  less sensitive to hyperparameters change.

**Summary Of The Review:**

While the practical viability of the proposed method is still uncertain, the insights on using softmax attention from this paper are valuable. The proposed synthetic evaluation framework is also a good foundation for further discussions on softmax-attention alternatives.

---

> ### Author Response · Authors · 2021-11-15
> **Reply to comments and questions**
>
> Thanks for your review and comments. To provide an answer to your concerns:
>
> 1. Please refer to our answer to reviewer fpkN for our reasoning why we did not (could not) include NLP evaluations. In particular, the core contribution of our paper is an investigation of the architectures, independent of a particular domain. Regarding our comment in Appendix C: We agree that experiments would be needed to confirm or deny any hypothesis here. However, our intuition is as follows: A softmax promotes sparse interactions. Many tasks in NLP can be efficiently solved by relying on a few key-words. We see this as a retrospective explanation of why softmax based models work so well in NLP tasks.
> 2. We agree, the NON architecture is a valid choice that performs very well in many setups. The benefits that a dynamic normalization like NAP brings are the following: (1) attention weights stay in a reasonable range, which helps the gradient propagation (partially addresses the vanishing/exploding gradient problems) and (2) if tokens are similar, their differences get amplified, helping the model by boosting relevant information. Empirical evidence (apart from Figure 7) for this can be seen especially in edge cases, e.g. NAP can handle higher data bias (Figure 6) or better find the mode in larger vocabulary sizes (Figure 25).
>
> We thank you also for the many suggestions. To provide answers:
>
> 1. We shortly implemented a weighted aggregation with learned aggregation weights akin to the Random Synthesizer in [1] but without the softmax. The results confirm our hypothesis, as the model performs quite well across a good range of hyperparameters in the setup of Figure 7. We also ran this model on the case-distinction task with outputs on all tokens and the mode finding task. It performs well in both. We will add these results in the revision.
> 2. One of the main reasons to move from BERT to MTE is to get rid of the focus on local information. We ran some further experiments in the last few days to see how NAP performs in a BERT setup (with learning rate warmup, gradients clipped, and layer normalization as in BERT). The results are complementary: the model is still more robust to hyperparameter changes, albeit not performing well on higher learning rates. On these high learning rates, the model focuses (similar to BERT) on the local ‘first’ task (due to the placement of layer normalization, which with the initialization of the mixing layer yields a focus on local information). We will add these results in the revision.
> 3. To be honest, we do not have a complete answer to these questions and we would love to see further research into this. Our current conjecture is the following: We do not believe that local information bias is beneficial here, since NAP and max also do not have a local information bias and perform better than BERT. Rather, the results might be explained by the gradient dynamics: Gradients in this setup come from one token and are broadcasted to the whole sequence. BERT starts off with a local focus and small gradient propagation to other tokens. The other models, on the other hand, start off with normalized gradients broadcasted to other tokens.  Max filters the gradient broadcasting and NAP normalizes the gradient broadcasting. MTE, NON and sum, however, can yield additive interference of the gradients which can result in unstable training dynamics. We agree that Figure 23 is more convincing to show what works and what does not in this setup. However, a main part of the paper is also about the focus on local information in relation to the training setup, which is better illustrated in Figure 7, which is why we opted for Figure 7 in the main text.
> 4. We agree. However, the page limit makes this close to impossible.
> 5. This is a good point. We will try to replicate the Figures with channels split and add them for reference at the end of the appendix.
>
>
> [1] Y. Tay et al., Synthesizer: Rethinking Self-Attention for Transformer Models

---

### Official Review · Reviewer_rjM7 · 2021-11-05

**Correctness:** 3
**Technical Novelty And Significance:** 3
**Empirical Novelty And Significance:** 4
**Recommendation:** 6
**Confidence:** 3

**Main Review:**

Strengths:

+   The paper is clearly motivated by the problem that current self-attention is not robust to hyperparameter changing. The authors also give some intuitive examples on this, such as bias towards local information and gradient vanishing.
+   The paper designs synthetic tasks and does a through analysis of NAP and other baseline models in this setting. Then the authors also experiment in some real-world settings such as GNN and RL.
+   The visualization method of the results is clever.
+   The experiment on finding the mode in the input data is inspiring, which implies that information is not effectively routing through all tokens in attention modules.
+   The paper is well-written and easy to follow.


Weaknesses:

-    A few related works are not cited or compared to. For example, [1] replace the softmax in self-attention with a token-wise Guassian kernel. [2] compares self-attention with fast weight programmers and proposes to remove the softmax. Several other works also adopt a similar design [3, 4]. Although these works are focusing on different problems instead of robustness to hyperparameter as in your paper, could you give some discussion on the relation between these prior arts and your model?
-    The authors did experiments in real-world settings (GNN and RL) in Sec 5.4, but the performance of NAP and BERT / MTE is somewhat similar. They all have the same validation results (99.1% in GNN, and 99.3%-99.4% in RL). The robustness to hyperparameters is also similar between MTE and NAP in RL task. Is it because the tasks are still too small in scale to show the difference between NAP and baseline models?
-    The normalization on $l^{i, j}_m$ will force the attention weights $a^{i, j}_m$ to have a fixed variance. But for example, if the tokens are all very similar, the attention weights should be similar too (thus having a small variance). Is the normalized attention able to handle this situation?


Other comments:

-    Some other works have also focused on stabilize the training process of transformers [5]. Do they have any relation to your work? A short discussion on this line of work could be beneficial.




[1] Lu J, Yao J, Zhang J, et al. SOFT: Softmax-free Transformer with Linear Complexity[J]. arXiv preprint arXiv:2110.11945, 2021.

[2] Schlag I, Irie K, Schmidhuber J. Linear transformers are secretly fast weight programmers[C]//International Conference on Machine
Learning. PMLR, 2021: 9355-9366.

[3] Shen Z, Zhang M, Zhao H, et al. Efficient attention: Attention with linear complexities[C]//Proceedings of the IEEE/CVF Winter Conference on Applications of Computer Vision. 2021: 3531-3539.

[4] Cao S. Choose a Transformer: Fourier or Galerkin[J]. arXiv preprint arXiv:2105.14995, 2021.

[5] Xiao T, Singh M, Mintun E, et al. Early convolutions help transformers see better[J]. arXiv preprint arXiv:2106.14881, 2021.

**Summary Of The Paper:**

This paper starts from the observation that current self-attention modules is sensitive to hyperparameter changing. The authors conjecture that this is due to the softmax operator in self-attention, and give some intuitive examples to support their hypothesis (eg., bias towards local information, gradient vanishing). To solve the problem, the authors propose to replace the softmax with a normalization operator. They show that normalized attention is more robust to hyperparameter in a synthetic dataset, as well as real-world settings.

**Summary Of The Review:**

The paper is well-motivated by the problem of sensitivity to hyperparameters and propose a sensible algorithm to solve the problem. The experiments on the synthetic task are sound. However, it seems lacking some convincing results on real-world dataset. And the relation to some prior works needs to be clarified.

---

> ### Author Response · Authors · 2021-11-15
> **Reply to comments and questions**
>
> Thank you for your review, comments and pointers. To answer your questions:
>
> _…could you give some discussion on the relation between these prior arts and your model?_
>
> We thank you for the pointer to these recent works, which (independent from us) provide complementary results that motivate architectures without softmax. In particular, [1] propose to use an exponentiated squared L2 distance between keys and queries as attention weights. Their main motivation is to then use a matrix decomposition to achieve linear complexity, noting that ``[...] without linearization, the training of a [kernel-based] transformer fails to converge’’.
> [2] focuses on the autoregressive setup, linking the head dimension d_h to a corresponding limitation in the memorization capacity of linearized models. Our work focuses on the encoder-only setup with bi-directional interactions where no such limitation exists. [3] propose an architecture similar to our NON architecture for improved computational complexity. However, in most of their experiments they still rely on a softmax for normalization. [4] also explores an architecture similar to our NON architecture and links it to a Petrov-Galerkin projection for operator approximation. They empirically find in an ablation that a layer normalization on the keys and queries separately improves convergence.
> Our work goes beyond this, showing the correlated effects between hyperparameters and architecture choices. In contrast to all mentioned works, we provide a viewpoint from the gradient dynamics perspective and investigate the effect of skewed data distributions.
> We will include this discussion in the revision of our work.
>
> _… Is it because the tasks are still too small in scale to show the difference between NAP and baseline models?_
>
> Our main argument here is to show that NAP performs as well as the baselines in these diverse setups. The tasks are actually not chosen in favor of NAP, as the average aggregation size (sequence length) is only 28.8 in the PPI task and ~6.7 in the RL task. For such a small aggregation size the bottleneck that a softmax introduces might not be that severe.
>
>
> _… if the tokens are all very similar, the attention weights should be similar too…_
>
> In this work, we challenge this view point. In particular, if tokens are similar, why shouldn’t we aim to amplify the differences that still make them distinct? As such, the model can potentially get more relevant information than just an approximately average token.
>
>
> _Some other works have also focused on stabilize the training process of transformers [5]. Do they have any relation to your work?_
>
> It is good to see that we are not the only people concerned with the stability of transformer training. However, the mentioned work [5] is not strongly related to our work and the findings are mostly orthogonal. In particular, [5] focuses on the vision domain and criticises the vision stem used before the Transformer blocks. In contrast, our work focuses on the Transformer architecture design itself.

---

> > ### Comment · Reviewer_rjM7 · 2021-11-25
> > **Thank the authors for the response**
> >
> > The response has partly resolved my concerns. I've raised my score to 6.

---

### Official Review · Reviewer_xpRQ · 2021-11-05

**Correctness:** 3
**Technical Novelty And Significance:** 3
**Empirical Novelty And Significance:** 2
**Recommendation:** 5
**Confidence:** 3

**Details Of Ethics Concerns:**

The work focuses on the topic of architectural module design, which is quite general rather than specializing to certain application tasks. But it would be helpful if the authors add some discussion in the paper.

**Main Review:**

Strengths:

The authors provide a theoretical analysis and proof on using softmax function in attention module for effect of gradient. I did not check the proof precisely. But analyzing neural network modules from a theoretical perspective is interesting and valuable to the community.

The whole paper is well organized and written, and easy to follow.

The authors give a comprehensive review and discussion to modern works related to transformer architectures and normalization and pooling methods.

The code is provided in the supplementary material, which would be helpful for reproducing the experiments in the paper.

Weaknesses:

The proposed NAP operation is quite related to layer normalization (LN). Instead learning element-wise scale and bias parameters in LN, scalar parameters are learned in NAP, which is supposed to function analogous to LN.  Can it be used as replacement of all the LN layers in the architecture?  If not considering variable length, is using LN operations only (i.e., replacing NAP as LN in the Figure 3) an effective variant compared to MTE and NAP architectures?

Most ablation studies and empirical analysis are conducted on synthetic tasks which are less convincing to demonstrate the effectiveness of the method. Although the authors present the experiments of exploring the proposed method into graph neural networks in PPI task and memory graph agent in reinforcement learning task, the performance improvements (in Fig. 9) are rather marginal compared to baseline methods.





**Summary Of The Paper:**

The authors investigate the self-attention module architecture and provide a theoretical analysis for the limitation of using softmax function for normalization in the module. In this regard, an unconstrained normalization function is proposed as alternative in the transformer. The authors compare it with baseline transformer and several variants to validate the effect of the proposed normalization function in transformer.

**Summary Of The Review:**

The authors theoretically analyze the limitation of using softmax function in attention module and propose an unconstrained surrogate in the paper. However, the empirical evidences shown in the paper are insufficient to validate the method.

---

> ### Author Response · Authors · 2021-11-15
> **Reply to comments and questions**
>
> Thank you for your review and comments. To answer your questions and concerns:
>
> _Can it be used as replacement of all the LN layers in the architecture?_
>
> The restriction to scalar parameters in our normalization operation is mainly done to allow for variable sequence lengths. Using scalar parameters in all layer normalizations would be possible, but allows for less flexibility. We therefore did not consider using scalar parameters in the layer normalizations, as the size of the hidden layers stay fixed.
>
> _If not considering variable length, is using LN operations only (i.e., replacing NAP as LN in the Figure 3) an effective variant compared to MTE and NAP architectures?_
>
> We are not entirely sure what the proposal is here. We discuss two interpretations, but please let us know if we misunderstood you:
> 1. Replacing the entire attention mechanism with LN, i.e., no projection to query, key and value and dot-product. This would reduce the information exchange between the tokens to the statistics (mean and standard deviation) that shift and scale all tokens equally. Using a typical LN, each token would then be scaled and shifted by a position-dependent parameter. This approach would share similarities with the sum and max architectures explored in our study, but would definitely also be interesting to explore.
> 2. Replacing the normalization in our NAP architecture with LN, that is, after the dot-product between keys and queries, normalize with a scale and shift parameter per token position. This would give a position-dependent weighting of value vectors. Also this would be interesting to explore if the sequence length stays fixed.
> If you are interested in a particular result of either of those two, please let us know. We should be able to run some further experiments during the revision period (a full run on a synthetic task takes 1-2 days for a single architecture).
>
> _Most ablation studies and empirical analysis are conducted on synthetic tasks which are less convincing [...]_
>
> Note that the main aim of the paper is to provide insights into architectural choices. For this, synthetic tasks are essential, as they allow us to vary the key aspects that we are interested in. We see this as an important step towards a better understanding of these architectures, as the single peak performance reported in most papers tells little about how the architecture might perform in a completely different domain. We therefore aimed for a neutral comparison of different choices in a large variety of setups instead of beating the state of the art in a particular domain. While it is by no doubt important to advance the state of the art performance in difficult benchmarks, we argue that there is a lot of value in taking a step back and evaluating architecture choices in isolation. In particular, the presented insights can help to decide, what kind of aggregation to use in a given setup, how to adjust hyperparameters if our model does not work, or where a bottleneck in the architecture might prevent efficient learning. Moreover, we present a method to investigate how architectures handle biases in the data. This can be of great importance if we seek to build models that adhere to ethical standards that might be misrepresented in the data.
>
> _(Ethical Concerns) [...] it would be helpful if the authors add some discussion in the paper._
>
> We will do so. In particular, we will add a discussion on how biases in the data might be linked to ethical concerns and the insights our results provide in that regard.

---

### Official Review · Reviewer_fpkN · 2021-11-07

**Correctness:** 3
**Technical Novelty And Significance:** 3
**Empirical Novelty And Significance:** 3
**Recommendation:** 6
**Confidence:** 3

**Main Review:**

The finding of the paper is interesting - the arguments made about the restrictive nature of the softmax operator makes sense. The proposal to replace the softmax with normalization is very simple and could have a lot of implications. The experiments conducted on the synthetic data are thorough and the discussions were made well. However, the authors could have shown gone beyond synthetic experiments and shown more results on real world datasets.

**Summary Of The Paper:**

This paper looks at one specific aspect of transformer architectures - the normalization step that constrains the attention vectors in a probability simplex. The authors argue that this restriction has some limitations including limited information flow, makes the models biased towards local information at initialization and increases the sensitive to hyperparameters. To mitigate this, the authors propose an architecture that replaces the softmax in the attention mask with a simple normalization. Then, with several experiments on a synthetic dataset, the authors demonstrate that the proposed architectural change leads to models that are robust to hyperparameters, and less sensitive to biases in the data.

**Summary Of The Review:**

While I enjoyed reading the paper, my biggest concern is lack of real world experiments. I appreciate the authors on conducting experiments on multiple runs, repeating the experiments with different model sizes, visualizing the results effectively and conducting carefully crafted simple experiments that illustrates the specific points being made. The carefully experiments were all designed well. However, for practitioners, a pressing question would be if these architectures can be used as a replacement of the traditional transformer models. Can this be used for training language models or benchmark NLP tasks?

To this end, it would have been really nice if authors had shown one experiment on real world datasets. I understand that some institutions might not have enough resources to train large-scale models, but a proof of concept at a decent scale on a realistic dataset would have been nice. Such an experiment would tell whether these findings would translate to large scale datasets and tasks. I would also like to point out that the authors have one experiment on protein-protein interaction graph prediction, but I am not aware of this benchmark and I am unable to gain much insights from this experiment aside from the fact that the model works.

While this issue is certainly concerning to me, I believe the findings in this paper are a reasonably good contribution that could help others in the community. It's just that the paper would have been really strong had the authors gone beyond synthetic dataset and done some benchmark NLP experiments.

---

> ### Author Response · Authors · 2021-11-15
> **On the lack of NLP benchmarks**
>
> Thank you for your thoughts and evaluations. We agree, an evaluation on NLP benchmarks would strengthen the paper. We expand here our reasoning why we did not (could not) include such an evaluation:
>
> 1. NLP models thrive from large data sets and unsupervised pre-training. Pre-training a decently large model for as many hyperparameter combinations as we do would be prohibitively expensive. While there are multiple small scale NLP data sets (e.g., IMdB, MNLI, …), these data sets on their own are too small to capture the intricacies of our language. Training any model on a small data set in isolation will yield results that are far from the state of the art and therefore not of much use to the community.
>
> 2. We focus in our work on the training dynamics of the architectures and aim to provide the findings independent of a particular downstream application. We therefore opted to eliminate as many confounding variables as possible. Any reasonable NLP application comes with a range of additional choices which alter the results and would have to be ablated individually (choice of tokenizer, pre-training objective, sampling method in generative tasks, regularization and other hyperparameter choices). While we could take the latest proposal in each of those, we fear that this would make our findings obsolete in a few years, as each of these choices are still actively researched.
>
> 3. With research coming out that questions the implementation details [1, 2], we aim to shift focus: Instead of tuning our architecture until it achieves a miniscule improvement in accuracy, we aim to provide insights on which architectural changes yield which effect. We see this as a more sustainable road for future improvements in architecture design.
>
> [1] R. Wightman et al., ResNet strikes back: An improved training procedure in timm
> [2] L. Engstorm et al., Implementation Matters in Deep RL: A Case Study on PPO and TRPO

---

### Author Response · Authors · 2021-11-18
**Revision**

We just uploaded a revised version of our work. What we changed:

 - updated related work
 - minor correction in a calculation mistake that was in the submitted version. Concerns end of section 3, (c>9.2 instead of c>8.5) and Appendix B.
 - new Figure 14 (NAP in BERT setup) with corresponding text at the end of App. D.1
 - new Appendix H (Learned Aggregation Weights)
 - new Appendix K (Broader Impact Statement)
 - new Appendix L (Figure Replication for Colorblind People)
 - minor rewrites to reference new appendices and remain within page limitation

We invite readers and reviewers to ask further questions if they have any - we are happy to answer.

---

### Decision · Program_Chairs · 2022-01-20

**Decision:**

Reject

**Comment:**

This paper studies potential drawbacks in using softmax over attention in Transformers and evaluates other normalization approaches. Reviewers, while had been positive about the empirical analysis and the insights from the synthetic data experiments, agree that the paper lacks real world experiments/insights. I agree with that and believe the paper falls short in several areas.

1) Drawbacks of softmax: Paper states several generic drawbacks of softmax such as saturation issue leading to vanishing gradients. However the paper does not demonstrate if Transformer models used in practice suffer from this issue under standard training settings. Even the arguments that attention layer focuses on local information is quite vague and not well supported. Overall the analysis is quite weak without much concrete statements and demonstration in real settings.

2) Experiments: The paper presents many synthetic experiments evaluating alternates to softmax varying from layer normalization to pooling. There are no experiments showing if the studied variations actually solve the issues discussed in earlier section. Finally due to the lack of any real world experiments (even small scale ones), it is not clear if the results apply in real world settings.

Overall I think the paper needs significant work in formalizing the drawbacks of using softmax in Transformers and demonstrating that the proposed solutions indeed solve this problem.